# Cavin3 released from caveolae interacts with BRCA1 to regulate the cellular stress response

Kerrie-Ann McMahon[1]\*, David A Stroud[2], Yann Gambin[1†], Vikas Tillu[1], Michele Bastiani[1], Emma Sierecki[1†], Mark E Polinkovsky[1‡], Thomas E Hall[1], Guillermo A Gomez[1§], Yeping Wu[1], Marie-Odile Parat[3], Nick Martel[1], Harriet P Lo[1], Kum Kum Khanna[4], Kirill Alexandrov[1#], Roger Daly[5], Alpha Yap[1], Michael T Ryan[5], Robert G Parton[1,6]\*

[1]Institute for Molecular Bioscience, The University of Queensland, Queensland, Australia; [2]Department of Biochemistry and Molecular Biology, Bio21 Molecular Science and Biotechnology Institute, The University of Melbourne, Parkville, Australia; [3]School of Pharmacy, The University of Queensland, Woolloongabba, Australia; [4]Signal Transduction Laboratory, QIMR Berghofer Medical Research Institute, Queensland, Australia; [5]Monash Biomedicine Discovery Institute, Department of Biochemistry & Molecular Biology, Monash University, Melbourne, Australia; [6]Centre for Microscopy and Microanalysis, The University of Queensland, Queensland, Australia

\*For correspondence:
k.mcmahon@imb.uq.edu.au (K-AMM);
r.parton@imb.uq.edu.au (RGP)

Present address: [†]EMBL Australia Node in Single Molecule Science, Lowy Cancer building, Level Medical Sciences UNSW Kensington Campus, Sydney, Australia; [‡]Acrondis AG, Christoph Merian-Ring, Reinach, Switzerland; [§]Centre for Cancer Biology, South Australia Pathology and the University of South Australia, Adelaide, Australia; [#]CSIRO-QUT Synthetic Biology Alliance, ARC Centre of Excellence in Synthetic Biology, Centre for Agriculture and the Bioeconomy, Institute of Health and Biomedical Innovation School of Biology and Environmental Science, Queensland University of Technology, Brisbane, Australia

Competing interests: The authors declare that no competing interests exist.

**Abstract** Caveolae-associated protein 3 (cavin3) is inactivated in most cancers. We characterized how cavin3 affects the cellular proteome using genome-edited cells together with label-free quantitative proteomics. These studies revealed a prominent role for cavin3 in DNA repair, with BRCA1 and BRCA1 A-complex components being downregulated on cavin3 deletion. Cellular and cell-free expression assays revealed a direct interaction between BRCA1 and cavin3 that occurs when cavin3 is released from caveolae that are disassembled in response to UV and mechanical stress. Overexpression and RNAi-depletion revealed that cavin3 sensitized various cancer cells to UV-induced apoptosis. Supporting a role in DNA repair, cavin3-deficient cells were sensitive to PARP inhibition, where concomitant depletion of 53BP1 restored BRCA1-dependent sensitivity to PARP inhibition. We conclude that cavin3 functions together with BRCA1 in multiple cancer-related pathways. The loss of cavin3 function may provide tumor cell survival by attenuating apoptotic sensitivity and hindering DNA repair under chronic stress conditions.

## Introduction

Caveolae are an abundant surface feature of most vertebrate cells. Morphologically, caveolae are 50–100 nm bulb-shaped structures attached to the plasma membrane (*Parton and del Pozo, 2013*). One of the defining features of this domain is the integral membrane protein caveolin-1 (CAV1). CAV1 is a structural component of caveolae regulating diverse cellular processes, including endocytosis, vesicular transport, cell migration, and signal transduction (*Parton and del Pozo, 2013*).

Recently, we and others have characterized a caveolar adaptor molecule, caveolae-associated protein 3 (cavin3) (*McMahon et al., 2009*). Cavin3 belongs to a family of proteins that includes caveolae-associated protein 1 (cavin1), caveolae-associated protein 2 (cavin2), and the muscle-specific member caveolae-associated protein 4 (cavin4) (*Ariotti and Parton, 2013*; *Bastiani et al., 2009*; *Hansen et al., 2009*; *Kovtun et al., 2015*; *Lo et al., 2015*; *McMahon et al., 2009*). Cavin3 is

**eLife digest** When cells become cancerous they often stop making certain proteins. This includes a protein known as cavin3 which resides in bulb-shaped pits of the membrane that surrounds the cell called caveolae. These structures work like stress detectors, picking up changes in the membrane and releasing proteins, such as cavin3, into the cell's interior.

Past studies suggest that cavin3 might interact with a protein called BRCA1 that suppresses the formation of tumors. Cells with mutations in the gene for BRCA1 struggle to fix damage in their DNA, and have to rely on other repair proteins, such as PARPs (short for poly (ADP-ribose) polymerases). Blocking PARP proteins with drugs can kill cancer cells with problems in their BRCA1 proteins. However, it was unclear what role cavin3 plays in this mechanism.

To investigate this, McMahon et al. exposed cells grown in the laboratory to DNA-damaging UV light to stimulate the release of cavin3 from caveolae. This revealed that cavin3 interacts with BRCA1 when cells are under stress, and helps stabilize the protein so it can perform DNA repairs. Cells without cavin3 showed decreased levels of the BRCA1 protein, but compensated for the loss of BRCA1 by increasing the levels of their PARP proteins. These cells also had increased DNA damage following treatment with drugs that block PARPs, similar to cancer cells carrying mutations in the gene for BRCA1.

These findings suggest that cavin3 helps BRCA1 to suppress the formation of tumors, and therefore should be considered when developing new anti-cancer treatments.

epigenetically silenced in a range of human malignancies (*Xu et al., 2001*), principally due to hyper-methylation of its promoter region (*Carén et al., 2011*; *Kim et al., 2014*; *Lee et al., 2008*; *Lee et al., 2011*; *Martinez et al., 2009*; *Tong et al., 2010*; *Zöchbauer-Müller et al., 2005*). Furthermore, cavin3 has been previously suggested to interact with BRCA1, although no data has been formally published to support this interaction (*Xu et al., 2001*). Several studies have implicated cavin3 in a broad range of cancer-related processes including proliferation, apoptosis, Warburg metabolism, as well as in cell migration and matrix metalloproteinase regulation; however, the molecular basis of its actions is poorly understood (*Hernandez et al., 2013*; *Toufaily et al., 2014*).

BReast CAncer gene 1 (BRCA1) is a significant breast cancer suppressor gene. It is one of the most frequently mutated genes in hereditary breast cancer (*King and Marks, 2003*; *Miki et al., 1994*; *Venkitaraman, 2002*). Also, BRCA1 levels are reduced or absent in many sporadic breast cancers due to gene silencing by promoter methylation or downregulation of the gene by other tumor suppressors or oncogenes (*Mueller and Roskelley, 2003*; *Turner et al., 2004*). BRCA1 has been implicated in a remarkable number of processes, including cell cycle checkpoint control, DNA damage repair, and transcriptional regulation (reviewed by *Lord and Ashworth, 2016*; *Savage and Harkin, 2015*). At the molecular level, accumulated evidence suggests that BRCA1 plays an integral role in the formation of several macromolecular complexes (BRCA1 A, BRCA1 B, and BRCA C, with different associated proteins) that participate in distinct processes to repair DNA damage (*Deng and Brodie, 2000*; *Huen et al., 2010*; *Roy et al., 2012*; *Scully et al., 1997*; *Scully et al., 1999*; *Scully and Livingston, 2000*; *Wang et al., 2007*).

Specifically, the BRCA1 A-complex consists of BRCA1 in association with RAP80, the deubiquitinating (DUB) enzymes BRCC36 and BRCC45, MERIT-40, and the adaptor protein ABRAXAS1 (*Harris and Khanna, 2011*; *Her et al., 2016*; *Savage and Harkin, 2015*; *Wang et al., 2007*). The BRCA1 A-complex participates in DNA repair by targeting BRCA1 to ionizing radiation (IR)-inducible foci; this occurs when RAP80 interacts with K63 poly-ubiquitin chains at sites of double strand breaks (DSBs) where the DNA damage marker γH2AX is phosphorylated (*Yan and Jetten, 2008*). BRCA1-A complex is thought to target BRCA1 to sites of DSB through interaction with ubiquitin-interacting motifs of RAP80, which recognize the Lys63 poly-ubiquitin chains of H2AX (*Sobhian et al., 2007*; *Wang et al., 2007*; *Yan and Jetten, 2008*). BRCA1 is also bound to BRCA1-associated Ring Domain 1 (BARD1), an interaction that is necessary for BRCA1 protein stability, nuclear localization, and E3 ubiquitin ligase activity (*Irminger-Finger et al., 2016*). In addition, BRCA1 is also a nuclear-cytoplasmic shuttling protein, and increasing evidence suggests that BRCA1 function can be controlled via active shuttling between subcellular compartments (*Fabbro et al., 2002*; *Feng et al., 2004*).

We identify a novel function for cavin3 mediated through its interaction with BRCA1, leading to regulation of BRCA1 levels, subcellular location, and function. We show that cavin3 controls BRCA1 functions in UV-induced apoptosis and cell protection against DNA damage through downregulated recruitment of the BRCA1 A-complex to DNA lesions in response to UV damage.

## Results

### Global proteome analyses of cavin3 function reveal a prominent role in DNA repair

As a first step to investigate the cell biology of cavin3, we undertook an unbiased approach to characterize its cellular proteome, using label-free quantitative (LFQ) proteomics. We deleted cavin3 by genome editing in HeLa cells, a well-characterized model system that has been used extensively to study caveolae (*Bohmer and Jordan, 2015*; *Boucrot et al., 2011*; *Hao et al., 2012*; *Hirama et al., 2017*; *Pang et al., 2004*; *Rejman et al., 2005*; *Sinha et al., 2011*; *Figure 1A*, *Figure 1—figure supplement 1A*). Global proteome analyses were carried out with three replicates from matched WT and cavin3 KO HeLa cells. Cells were SILAC-labeled and subjected to mass spectrometric analysis after lysis. Relative protein expression differences were then determined using label-free quantitation (*Figure 1A*). A total of 4206 proteins were robustly quantified with >2 unique peptides and an FDR < 1.0% in at least two out of three replicates (*Figure 1A*, details in *Supplementary file 1*). To validate these results, we immunoblotted for several proteins involved in diverse cellular processes. Levels of these proteins were consistent with the proteomic analysis (*Figure 1—figure supplement 1B*). Their levels were restored by the expression of exogenous cavin3, confirming the specificity of the KO effect (*Figure 1—figure supplement 1C*).

Our analysis revealed distinct cavin3-dependent protein networks that might yield new insights into its cellular function. Initial inspection of differentially expressed protein by Gene Ontology analysis revealed that many proteins involved in DNA repair were altered in cavin3 KO cells (*Figure 1B, C* and *Supplementary file 2*); see *Supplementary file 3* for further analysis of cavin3-dependent pathways. Strikingly, BRCA1 (~1.5-fold decrease) and many components of the BRCA1 A-complex, BRCC36 (~1.5-fold decrease), MDC1 (~1.7-fold decrease), and the newly described UBE4A (~2.2-fold decrease, *Baranes-Bachar et al., 2018*), were reduced in cavin3 KO cells that were confirmed by western analysis (*Figure 1D, Figure 1—figure supplement 1D*). In contrast, 53BP1 protein levels were increased in cavin3 KO cells (*Figure 1D, Figure 1—figure supplement 1D*). Accordingly, we elected to pursue the relationship between cavin3 and BRCA1 in greater detail.

### Cavin3 interacts with BRCA1 in vitro and a model cell system

First, we asked whether cavin3 and BRCA1 might interact in the cytosol. Recent studies suggest that the release of cavin proteins into the cytosol can allow interaction with intracellular targets (*Gambin et al., 2014*; *McMahon et al., 2019*; *Sinha et al., 2011*). To test whether non-caveolar cavin interacts with BRCA1, we used MCF7 cells as a model system. These cells lack endogenous CAV1, cavins, and caveolae (*Gambin et al., 2014*; *McMahon et al., 2019*), and so expressed cavin proteins are predominantly cytosolic.

BRCA1-GFP was coexpressed in MCF7 cells with exogenous mCherry-tagged cavins-1, 2, 3, and mCherry-CAV1, and interactions between these proteins were measured in cytoplasmic extracts using two-color single-molecule coincidence (SMC) detection. The numbers of photons detected in green and red channels were plotted as a function of time where each fluorescent burst was analyzed for the coincidence between the GFP and cherry fluorescence that reflects co-diffusion of at least two proteins with different tags, the total brightness of the burst, indicating the number of proteins present in the oligomer and the burst profile that is determined by the rate of diffusion and reflects the apparent size of the complex (*Gambin et al., 2014*). This revealed a specific association between BRCA1 and cavin3-mCherry, but not with the other cavin proteins (*Figure 2A–E*). Quantitatively, 60% of BRCA1-GFP associated with cavin3-mCherry (*Figure 2D*). The distribution of bursts revealed the behavior of monomeric GFP. This data was used to calibrate the brightness profile and estimate the number of BRCA1-GFP molecules. We concluded that overexpressed BRCA1 primarily exists in a dimeric state when expressed in MCF7 cells and that a dimer of overexpressed BRCA1 interacts with a monomer of exogenous cavin3 (*Figure 2F*). Similar results were obtained when

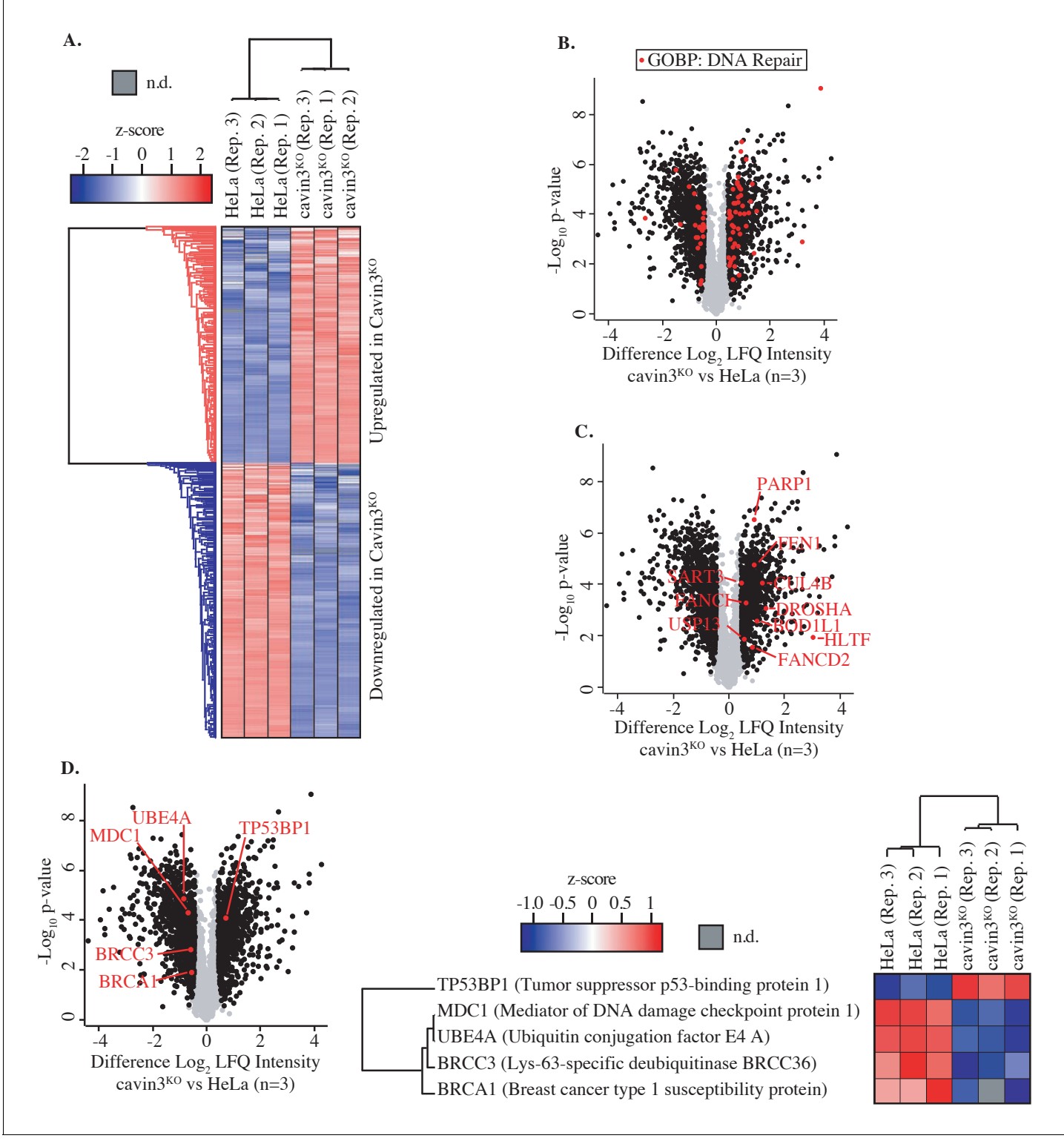

**Figure 1.** Global proteome analysis of cavin3 KO HeLa cells by label-free quantitative proteomics. (A) Z-score for HeLa WT and cavin3 KO cells (replicates Rep. 1–3) showing upregulated proteins (red) and downregulated proteins (blue). (B) Volcano plot showing proteins (red dots) identified by Gene Ontology Biological Process (GOBP) involved in DNA repair. (C) Volcano plot showing DNA repair proteins upregulated in cavin3 KO cells. (D) Volcano plot showing proteins of the BRCA1 A-complex, BRCA1, BRCC3, MDC1, and UBE4A downregulated in cavin3 HeLa KO cells and upregulation of 53BP1 with a heatmap analysis of the expression of each of these proteins in replicate (Rep. 1–3) HeLa WT and cavin3 KO cells.

The online version of this article includes the following source data and figure supplement(s) for figure 1:

*Figure 1 continued on next page*

Figure 1 continued

**Figure supplement 1.** General characterization of cavin3 KO HeLa cells.
**Figure supplement 1—source data 1.** Raw western data for HeLa WT and cavin3 KO cells with molecular weight markers for *Figure 1—figure supplement 1B*.
**Figure supplement 1—source data 2.** Raw western data for HeLa WT, cavin3 KO, and cavin3 KO with cavin3-GFP cells with molecular weight markers for *Figure 1—figure supplement 1C*.
**Figure supplement 1—source data 3.** Raw western data for HeLa WT, cavin3 KO, and cavin3 KO with cavin3-GFP cells with molecular weight markers for *Figure 1—figure supplement 1C*.

BRCA1-GFP and cavin3-mCherry were coexpressed in MDA-MB231 cells, a cell line with endogenous caveolar proteins and abundant caveolae at the plasma membrane (*Figure 2—figure supplement 1A–E*). These findings implied that BRCA1 and cavin3 can interact in the cytosol, irrespective of the cells' caveolar state.

We then used a *Leishmania* cell-free system (*Gambin et al., 2014*; *Sierecki et al., 2013*) to test whether these proteins can interact directly. Indeed, a construct bearing the first 300 amino acids of BRCA1 (1–300, tr-BRCA1), which contains the nuclear export signal (NES) and BARD1 binding sites (*Figure 2G*), was associated with cavin3 (*Figure 2J*), but not with the other cavin proteins (*Figure 2H, I*). These data suggest that cavin3 directly binds to the N-terminus of BRCA1.

Finally, we used in situ proximity ligation assay (PLA) technology (*Söderberg et al., 2007*) to probe for the protein-protein association within intact cells. GFP-tagged cavins or CAV1-GFP were expressed in MCF7 cells, and potential associations between transgenes and endogenous BRCA1 were analyzed using anti-BRCA1 and anti-GFP antibodies. Positive interactions in PLA analyses are revealed by fluorescent puncta (*Figure 3A–E*). Puncta were evident throughout the cytosol of cells expressing cavin3-GFP, but not with the other cavins, CAV1-GFP or GFP alone (*Figure 3A–E*, quantitation in *Figure 3F*). Additional experiments using different combinations of antibodies (e.g., rabbit antibodies against endogenous BRCA1 together with mouse anti-GFP antibodies; *Figure 3—figure supplement 1*) yielded similar results. Control experiments (GFP alone, BRCA1 alone, absence of PLA probes, and no antibody) yielded few puncta (*Figure 3—figure supplement 2A–E*). Collectively, these studies suggest that BRCA1 can interact with cavin3 directly *in vitro* and that expressed cavin3 can associate with endogenous BRCA1 in cells.

## Cavin3 regulates BRCA1 protein expression and localization

We next examined the relationship between cavin3 and the subcellular localization of BRCA1. Immunofluorescence revealed a typical nuclear staining pattern for endogenous BRCA1 with little cytoplasmic staining in control MCF7 cells and cells expressing cavin1-GFP (*Figure 4A*). In contrast, the expression of cavin3-GFP increased cytosolic staining for endogenous BRCA1 (*Figure 4A*), and this was confirmed by quantitative analysis of the protein distribution (*Figure 4B*). Western blotting revealed that cavin3-GFP selective increased total cellular levels of BRCA1 (*Figure 4C*, quantitation in *Figure 4—figure supplement 1A*). This represents a post-transcriptional effect of cavin3 as BRCA1 mRNA levels were not significantly increased (*Figure 4—figure supplement 1B*). Interestingly, the proteasome inhibitor, MG132, increased BRCA1 levels in control cells, consistent with evidence for proteasomal degradation of BRCA1 (*Choudhury et al., 2004*). However, it did not increase the already-elevated levels of BRCA1 found in cavin3-GFP cells (*Figure 4D*, quantitation in *Figure 4—figure supplement 1C*).

Dependence of BRCA1 on cavin3 was also evident when cavin3 was depleted in either A431 and MDA-MB231 cells, using two different siRNAs (*Figure 4E*, quantitation in *Figure 4—figure supplement 1D*, *Figure 4—figure supplement 2A, C*). These cell lines express cavin3, CAV1, and BRCA1 proteins and present caveolae at the plasma membrane (*Figure 4—figure supplement 1E*). In both cases, cavin3 depletion caused a significant decrease in BRCA1 (*Figure 4E*, quantitation in *Figure 4—figure supplement 1D*, *Figure 4—figure supplement 2A, C*), and this was abrogated by proteasome inhibition (*Figure 4G*). Immunofluorescence staining revealed that BRCA1 was reduced in the cytosol and nuclei of cavin3 siRNA cells (*Figure 4—figure supplement 3*). Interestingly, depletion of BRCA1 with two independent siRNAs significantly decreased endogenous cavin3 protein levels in these cells (*Figure 4F*, quantitation in *Figure 4—figure supplement 1F*, *Figure 4—figure*

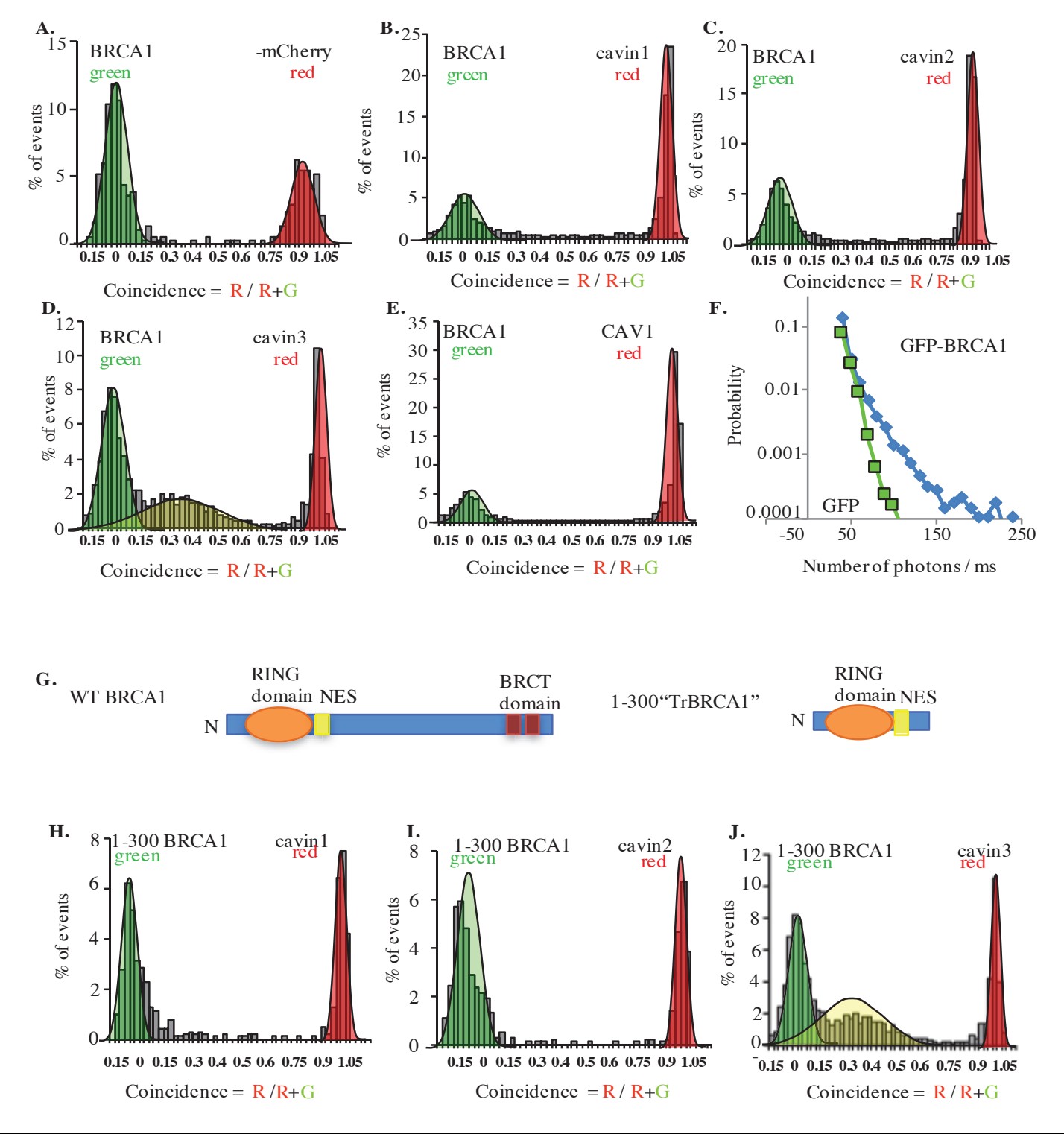

**Figure 2.** Single-molecule analysis of BRCA1 with cavin3-mCherry in MCF7 cells. (A) Two-color single molecule fluorescence coincidence of BRCA1-GFP with (A) mCherry control, (B) mCherry-cavin1, (C) mCherry-cavin2, (D) mCherry-cavin3, and (E) mCherry-CAV1 coexpressed in MCF7 cells. The green curve represents BRCA1-GFP-only events, the red curve represents mCherry-only events, and the yellow curve represents BRCA1-GFP + Cherry events. (F) Distribution of burst brightness measured for BRCA1-GFP (blue) and GFP control (green). (G) Schematic representation of domain organization of full-length wildtype (WT) BRCA1 and the truncated (Tr) 1–300 BRCA1 constructs. NES: nuclear export signal; BRCT domain: BRCA1 C terminus domain;

*Figure 2 continued on next page*

Figure 2 continued

N: N terminus. (**H–J**). Two-color single-molecule fluorescence coincidence of 1–300 BRCA1 with (**H**) cavin1, (**I**) cavin2, and (**J**) cavin3 expressed in *Leishmania* cell-free lysates. More than 1000 events were collected in all cases.

The online version of this article includes the following figure supplement(s) for figure 2:

**Figure supplement 1.** Single-molecule analysis in MDA-MB231 cells.

---

*supplement 2B, D*). Taken with our earlier work on HeLa cells, these results collectively show that cavin3 can support BRCA1 protein levels in a variety of cancer cell systems.

## Cavin3 associates with BRCA1 when caveolae disassemble

What might induce cavin3 to interact with BRCA1? A variety of stresses cause caveolae to flatten and disassemble, releasing cavins into the cytosol. We, therefore, hypothesized that stimuli that induce caveola disassembly might induce the association of cavin3 with BRCA1.

First, we tested a role for mechanical stress by swelling cells with hypo-osmotic medium. We used A431 cells for these experiments as they have abundant caveolae. The total association between endogenous cavin3 and endogenous BRCA1, and their association in the nucleus, was significantly increased by hypo-osmotic stimulation, as measured by PLA (*Figure 5A*). No interaction was seen with a range of control proteins, including the nuclear proteins PCNA, flottilin1 and Aurora kinase (*Figure 5B–E*). These findings suggested that mechanical disassembly of caveolae could promote the association of cavin3 with BRCA1 both in the cytosol and the nucleus.

Nest, we tested the effect of non-mechanical stimuli by exposing cells to either UV (2 min pulse, 30 min chase) or oxidative stress with hydrogen peroxide ($H_2O_2$, 200 µM, 30 min). PLA showed that the interaction between endogenous BRCA1 and cavin3 was increased by both these stimuli (*Figure 6A–D*, top panel, quantitation in *Figure 6E*). A more extended time course further demonstrated that association between these proteins was evident at 30 min and maintained at low levels for up to 4 hr (*Figure 6F, G*). Interestingly, this coincided with a decrease in the interaction between cavin3 and cavin1, which occurs in caveolae (*Figure 6A–D*, bottom panel, quantitation in *Figure 6G*). Similar effects were seen in MDA-MB231 cells (*Figure 6—figure supplement 1*). Control experiments (knockdown of cavin3 or BRCA1 in untreated and UV-treated A431 cells) yielded few puncta (*Figure 6—figure supplement 2*), consistent with the notion that cavin3 was moving from caveolae into the cytosol to interact with BRCA1. Our findings indicate that cavin3 can be released to interact with BRCA1 when caveolae disassemble in response to various mechanical and non-mechanical stimuli.

## Cavin3 and BRCA1 function similarly in apoptosis in the cytosol and DNA damage sensing in the nucleus

Next, we sought to evaluate the potential functional consequences of this stress-inducible association of cavin3 with BRCA1. As cytoplasmic BRCA1 has been implicated in cell death pathways (*Dizin et al., 2008*; *Thangaraju et al., 2000*; *Wang et al., 2010*), we asked if cavin3 affects the sensitivity of cells to apoptosis induced by UV exposure. We found that LDH release, used as an index of membrane damage, was consistently increased after 2 min UV exposure in MCF7 cells that overexpressed cavin3-GFP, but not with cavin1-GFP (*Figure 7A*). This cell damage reflected apoptosis induction confirmed by staining for annexin V (which marks early apoptosis, *Figure 7B*) and the DNA dye 7-amino-actinomycin 7 (7-AAD, late apoptosis, *Figure 7C*). Both apoptotic markers were enhanced by cavin3-GFP overexpression. Thus, cavin3 could sensitize MCF7 cells to UV-induced apoptosis.

We then asked whether this effect also operated in cancer cells with endogenous expression of cavin3. Indeed, overexpression of cavin3-GFP significantly increased LDH release from UV-treated A431 and MDA-MB231 cells (*Figure 7D, F*). Furthermore, depletion of endogenous cavin3 reduced LDH release from these cells after UV stimulation (*Figure 7E, G*, controls in *Figure 7—figure supplement 1A, B*). Together, these findings indicate that cavin3 sensitizes cells to apoptosis induced by UV.

BRCA1 also sensitized A431 and MDA-MB231 cells to apoptosis, as evident when exogenous BRCA1 was overexpressed or the endogenous protein was depleted (*Figure 7E, G*, controls in

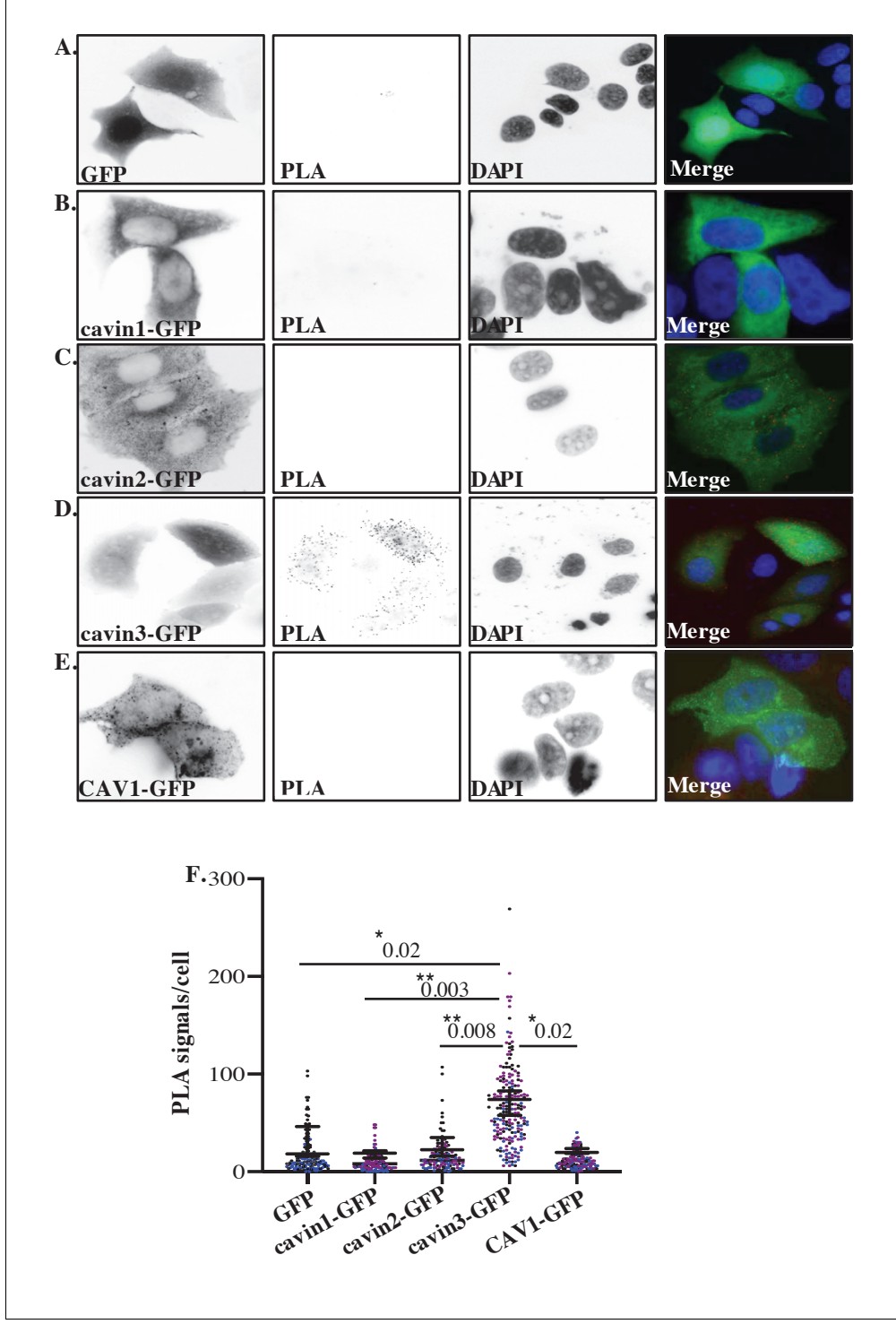

**Figure 3.** Proximity ligation assay (PLA) analysis of cavin3 and BRCA1 interaction in MCF7 cells. (A–E) Immunofluorescence microscopy in combination with PLA for protein-protein interactions (red dots) within single cells of stably expressing (**A**) MCF7-GFP, (**B**) MCF7-cavin1-GFP, (**C**) MCF7-cavin2-GFP, (**D**) MCF7-cavin3-GFP, and (**E**) MCF7-CAV1-GFP using monoclonal GFP and polyclonal BRCA1 antibodies. DNA was counterstained with DAPI (blue). Scale bars represent 10 µm. (**F**). Number of red dots/PLA signals in 40–50 cells for each MCF7-GFP-expressing cell line was quantified from three independent experiments using a nested ANOVA. Each biological replicate is color-coded, and the mean ± SEM is presented as a black bar. **p<0.05, **p<0.01.

The online version of this article includes the following figure supplement(s) for figure 3:

*Figure 3 continued on next page*

*Figure 3 continued*

**Figure supplement 1.** Proximity ligation assay (PLA) demonstrates cavin3 and BRCA1 interaction in MCF7 cells.
**Figure supplement 2.** Proximity ligation assay (PLA) controls.

*Figure 7—figure supplement 1C, D*). Therefore, we further examined the relationship between BRCA1 and cavin3. Overexpression of BRCA1 in cavin3-depleted A431 or MDA-MB231 cells or over-expression of cavin3 in BRCA1-depleted cells restored UV-induced apoptosis to control levels. This indicated that these two proteins have a similar sensitizing effect on UV-induced apoptosis (*Figure 7E, G*). These results suggest a pro-apoptotic role for both cavin3 and BRCA1 in stress-induced cancer cells. Similarly, in MCF7 cells expression of cavin3 alone or in combination with BRCA1 restored the sensitivity of BRCA1 KD cells to UV-induced apoptosis (*Figure 7—figure supplement 1E*). We further exposed WT and cavin3 KO HeLa cells to a range of stresses that allow interaction with BRCA1, including hypo-osmotic medium, UV, and oxidative stress (*Figure 7—figure supplement 2A–D*). Cavin3 KO cells exhibited enhanced resistance to all stressors, and apart from oxidative stress, this was time-dependent (*Figure 7—figure supplement 2A–D*). Overall, these findings suggest that BRCA1 and cavin3 participate together in the cellular stress response.

## Cavin3 protects against stress-induced DNA damage

In addition to promoting apoptosis, BRCA1, notably via its BRCA1 A-complex, has also been implicated in DNA repair to limit the mutational risk in stressed cells that evade apoptosis. As noted earlier, we found that BRCA1 A-complex components were reduced at steady state in cavin3 KO HeLa cells (*Figure 1A*). Next, we examined UV treatment on the level of these components in WT and cavin3 KO HeLa cells. As shown in *Figure 8A–C*, UV treatment of WT cells upregulated the expression of cavin3, BRCA1, the DNA damage marker, RAD51, and the A-complex proteins MDC1, Rap80, RNF168, and Merit40. Strikingly, the upregulation of BRCA1, RAD51, and the BRCA1 A-complex proteins was dramatically reduced in cavin3 KO cells (*Figure 8A, C*, quantitation in *Figure 8—figure supplement 1*). This suggested that cavin3 can influence the ability of BRCA1 to repair damaged DNA.

To test this, we first examined the response of BRCA1 to DNA damage. BRCA1 relocates to form foci at sites of DNA DSBs. Indeed, we found that BRCA1 foci increased within 30 min of UV irradiation (*Figure 8D, E*); however, this was significantly reduced in cavin3 KO cells (*Figure 8E*). Similarly, the recruitment of RAP80 and γH2AX was reduced in cavin3 KO cells, suggesting that DNA repair might be fundamentally compromised in these cells (*Figure 8E*).

Previous studies have shown that loss of functional BRCA1 protein leads to defects in DSB repair by homologous recombination and renders cells hypersensitive to PARP inhibitors through the mechanism of synthetic lethality (*Ashworth, 2008*; *Bryant et al., 2005*; *Farmer et al., 2005*; *Helleday et al., 2005*). Therefore, we asked whether cavin3 KO cells that are BRCA1 deficient are also sensitive to the PARP inhibitor, AZD2461.

Clonogenic survival assays and cell viability studies revealed that cavin3-deficient HeLa cells (red dots) were more sensitive to the PARP inhibitor AZD2461 at nM concentrations than control WT HeLa cells (black dots, *Figure 8—figure supplement 2*). As another means to look at PARP loss, WT and cavin3 HeLa KO cells were also depleted of PARP1 using CRISPR/Cas9 genome editing. Cavin3 and PARP1 KO cells failed to produce colonies in clonogenic survival assays with reduced cell viability (pink dots, *Figure 8—figure supplement 2*). These findings suggest that cavin3-deficient HeLa cells are sensitive to PARP inhibition, suggesting that cavin3 and BRCA1 are involved in homologous recombination repair. Furthermore, these findings suggest that PARP1 is a potential synthetic lethal partner for cavin3. We evaluated DNA strand breaks in control and PARP-treated WT HeLa and cavin3 KO cells using a comet assay, which revealed increased DNA damage only in PARP-treated cavin3 KO cells following a 6-day treatment (*Figure 8F*).

Recent reports have linked 53BP1 loss to PARP inhibitor resistance, presumably, as loss of 53BP1 partially restores homologous recombination repair in BRCA1-deficient cells (*Bouwman et al., 2010*; *Bunting et al., 2009*; *Cao et al., 2009*; *Turner et al., 2007*; *Yang et al., 2017*). This restoration is made possible because homologous recombination and non-homologous end-joining repair pathways compete to repair DNA breaks during DNA replication. Therefore, we determined the

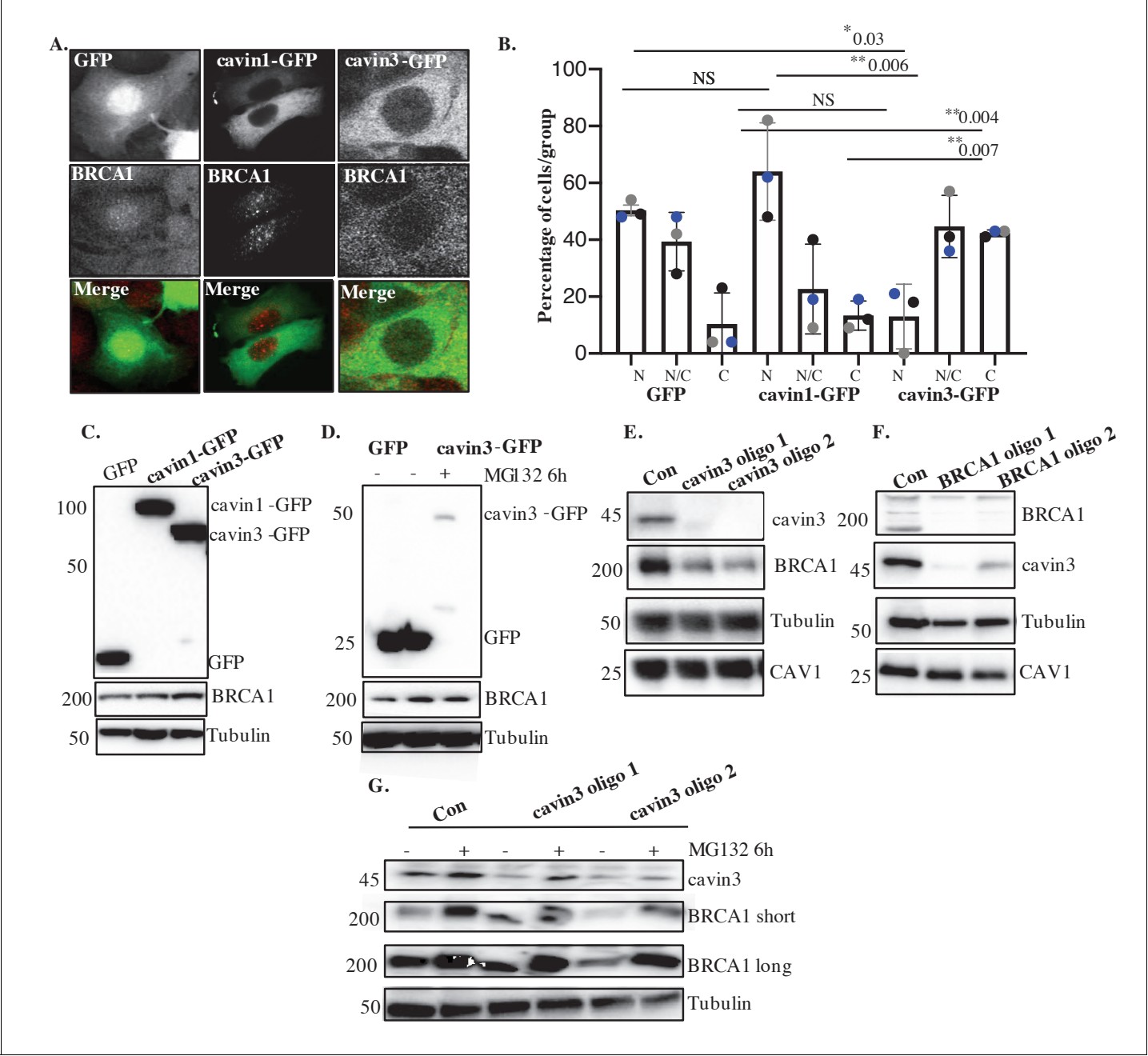

**Figure 4.** Cavin3 regulates BRCA1 protein expression and localization. (**A**) Representative image of MCF7 cells stably expressing GFP alone, cavin1-GFP, and cavin3-GFP fixed and stained with a BRCA1 antibody. (**B**) Percentage of MCF7 cells showing strictly nuclear, nuclear-cytoplasmic, or cytoplasmic localization of BRCA1 was counted for 50 cells from 4 to 5 independent experiments as mean ± SD using a one-way ANOVA and Bonferroni's multiple comparisons test. Each biological replicate was color-coded. NS: not significant, *p<0.05, **p<0.01. (**C**) Lysates from stably expressing MCF7 cells western blotted for GFP, BRCA1, and Tubulin as a load control. (**D**) MCF7-GFP and MCF7-cavin3-GFP cells, untreated (-) or treated with MG-132 for 6 hr. Lysates were western blotted with GFP, BRCA1, and Tubulin antibodies as a loading control. (**E**) A431 cells treated with control siRNAs (Con) or two siRNAs specific to cavin3. Lysates were western blotted using cavin3, BRCA1, CAV1 antibodies, and Tubulin as the loading control. (**F**) A431 cells treated with control siRNAs or two siRNAs specific to BRCA1. Lysates were western blotted using cavin3, BRCA1, CAV1 antibodies, and Tubulin as the loading control. (**G**) A431 cells treated with control (Con) or siRNAs specific to cavin3, untreated or treated with MG132 for 6 hr. Lysates were western blotted using cavin3, BRCA1, and Tubulin as a loading control. Quantitation of all blots in *Figure 4* is provided in *Figure 4—figure supplement 1A–E*.

The online version of this article includes the following source data and figure supplement(s) for figure 4:

**Source data 1.** Raw western data for MCF7 cells with molecular weight markers for *Figure 4C*.

*Figure 4 continued on next page*

*Figure 4 continued*

**Source data 2.** Raw western data for MCF7 cells with molecular weight markers for *Figure 4D*.
**Source data 3.** Raw western data for A431 cells with molecular weight markers for *Figure 4E*.
**Source data 4.** Raw western data for A431 cells with molecular weight markers for *Figure 4F*.
**Source data 5.** Raw western data for A431 cells with molecular weight markers for *Figure 4G*.
**Figure supplement 1.** Reciprocal regulation of BRCA1 and cavin3 protein levels.
**Figure supplement 1—source data 1.** Raw western data for HeLa WT and cavin3 KO cells with molecular weight markers for *Figure 4—figure supplement 1E*.
**Figure supplement 2.** Validation of loss of cavin3 and BRCA1 in MDA-MB231 cells.
**Figure supplement 2—source data 1.** Raw western data for MDA-MB231 cells with molecular weight markers for *Figure 4—figure supplement 2A*.
**Figure supplement 2—source data 2.** Raw western data for MDA-MB231 cells with molecular weight markers for *Figure 4—figure supplement 2B*.
**Figure supplement 3.** Reciprocal loss of BRCA1 and cavin3 in A431 cells.

dependence of the physiological outcomes on BRCA1 in cavin3 KO cells by rescue experiments with concomitant knockout of 53BP1. Loss of 53BP1 in cavin3 HeLa KO cells could revert the PARP sensitivity of these cells to WT cell levels as demonstrated in clonogenic survival and cell viability assays (orange dots, *Figure 8—figure supplement 2A–C*). These findings agree with several studies demonstrating that homologous recombination DNA repair is partially restored in BRCA1-deficient cells following 53BP1 loss (*Bouwman et al., 2010*; *Bunting et al., 2009*; *Cao et al., 2009*; *Turner et al., 2007*; *Yang et al., 2017*).

We further evaluated several other proteins: chromodomain helicase DNA-containing protein 3 (CHD3, an epigenetic modulator) and Fanconi anemia (FA) complementation Group 2 (FANCD2, a DNA damage sensor protein) that were specifically upregulated in cavin3 KO cells and that are involved in different aspects of DNA repair. These proteins represent potential targets and mediators of synthetic lethality in cancers (*Burdak-Rothkamm and Rothkamm, 2021*). Deficiencies in homologous recombination have been ascribed to cells with defects in several members of the FA pathway, including FANCD2 (*Ceccaldi et al., 2016*; *Jenkins et al., 2012*; *McCabe et al., 2006*; *Ridpath et al., 2007*); hence, we examined whether FANCD2-depleted cavin3 KO cells were sensitive to PARP inhibition. CHD3 is a chromatin remodeler related to CHD4, which is implicated as a tumor suppressor in several female malignancies (*Li and Mills, 2014*). It has been demonstrated that CHD3 can function like CHD4 in the nucleosome-remodeling (NuRD) complex and acts in the DNA damage response in active recruit of DNA repair factors to sites of lesions to promotion DNA repair (*Hoffmeister et al., 2017*; *Smith et al., 2018*).

Both CHD3 and FANCD2 were depleted in HeLa WT and cavin3 KO cells. Depletion of CHD3 and FANCD2 specifically in cavin3 KO cells induced profound cellular sensitivity to PARP inhibition in clonogenic survival and cell viability assays (*Figure 8—figure supplement 2A–C*). These findings suggest that cavin3 KO cells represent a novel cellular system to begin to dissect the interactions that occur in the DNA damage response, compensated that may occur by other components in a similar or different pathway for cell survival, and how this information can be used to identify new drug agents and treatment strategies in cancer.

## Discussion

Here we describe a novel role for caveolae and the cavin3 protein in regulating the critical tumor suppressor, BRCA1. Our studies raise the intriguing possibility that by releasing cavins, which can be triggered by mechanical and non-mechanical stimuli such as UV and oxidative stress (*McMahon et al., 2019*, and this study), caveolae can act as general sensors and transducers of cellular stress. Our findings suggest that defining the role of the cavin proteins may provide new insights into the functions of caveolae in pathological conditions such as cancer. Cavin3 may represent a promising therapeutic target in breast cancer through its ability to act both inside and outside of caveolae, by modulating specific signaling pathways (*Hernandez et al., 2013*) and by interacting with and modulating the expression of many proteins such as BRCA1, as shown here, and PP1alpha as previously described (*McMahon et al., 2019*).

The possibility of an interaction between BRCA1 and cavin3 was first suggested some 20 years ago, yet, no experimental evidence to support this interaction has been published to date (*Xu et al.,*

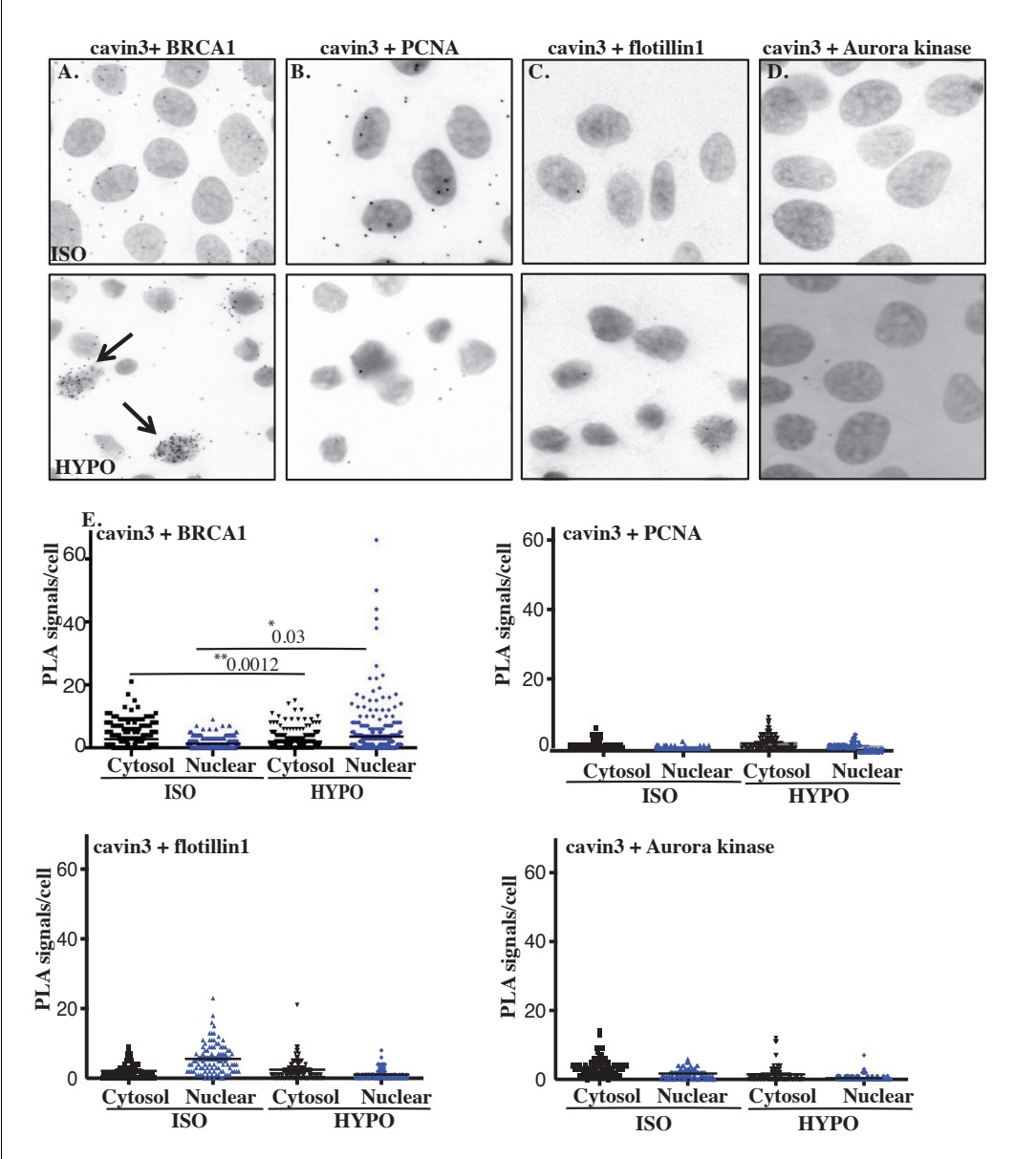

**Figure 5.** Cellular swelling of A431 cell causes an increase in the BRCA1-cavin3 interaction. (A) A431 cells were treated with isotonic (ISO) or hypo-osmotic (HYPO) medium, and proximity ligation assay (PLA) was performed using cavin3 and BRCA1, (B) cavin3 and PCNA, (C) cavin3 and flotillin1, and (D) cavin3 and Aurora kinase antibodies as controls for PLA. DNA was counterstained with DAPI (blue). Scale bars represent 10 μm. (E) Total number of PLA signals in the cytosol and the nucleus of cells as defined by DAPI staining in 50 cells for each pair of antibodies quantified from three independent experiments using a nested ANOVA with the mean ± SEM represented by the black bar, *p<0.05, **p<0.01.

2001). Our results provide the first clear evidence that cavin3 directly interacts with BRCA1 and that this occurs when cavin3 is released from caveolae in response to cellular stressors. We established this using multiple techniques, including PLA in MCF7, MDA-MB231 and A431 cells, SMC detection in multiple cancer cell lines (MCF7 and MDA-MB231 cells), and in vitro synthesized BRCA1 and cavin3. We were not able to reproducibly coimmunoprecipitate BRCA1 and cavin3. However, this technique can fail to detect weak or transient interactions (*Berggård et al., 2007*). Instead, the combination of cell-based methods (PLA and single-molecule approaches) and a cell-free direct interaction approach, as used here, provides unequivocal evidence for the proposed interaction between the N-terminus of BRCA1 and cavin3.

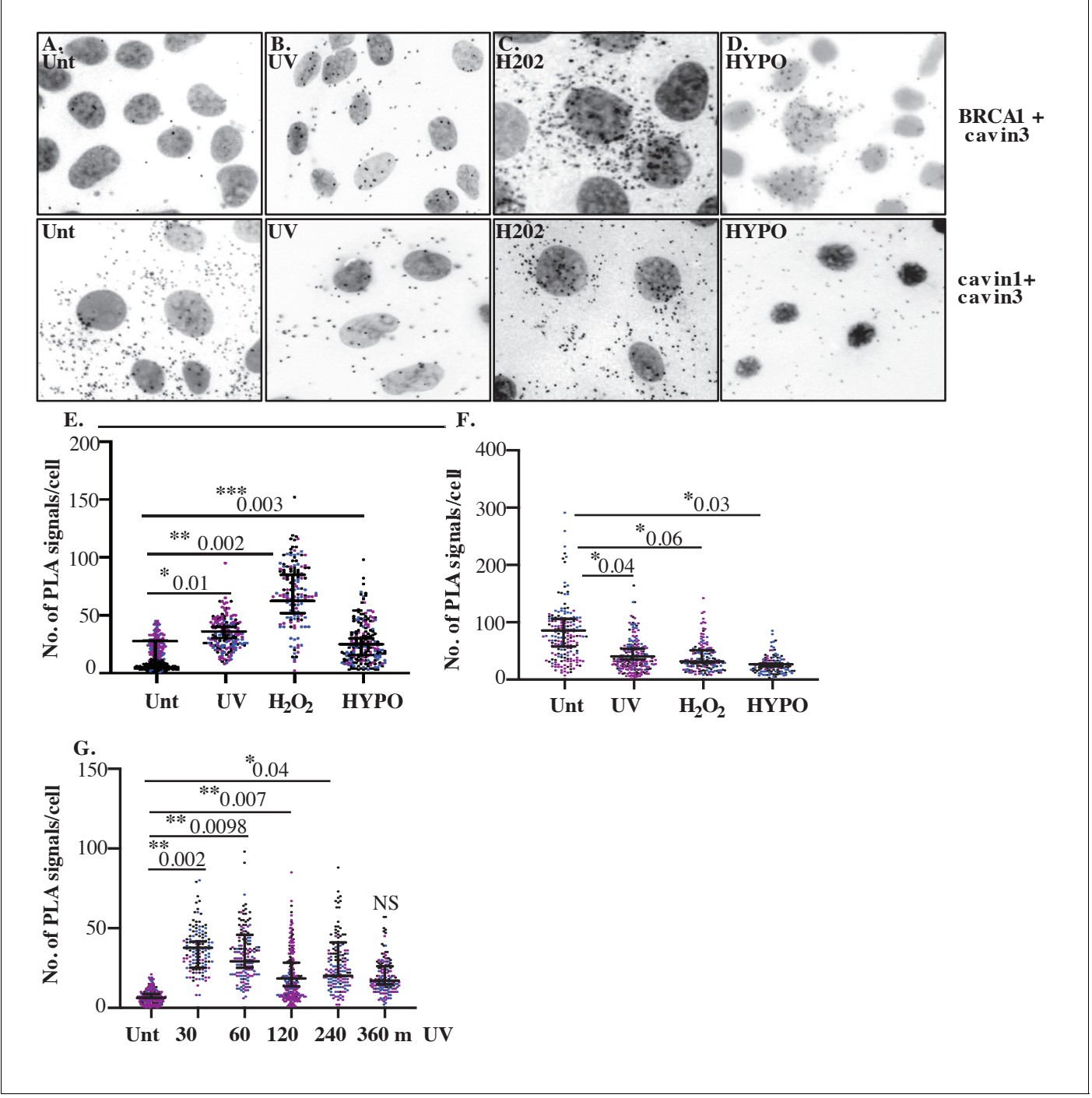

**Figure 6.** Close association between cavin3 and BRCA1 in A431 cells after stress treatment. (A) Immunofluorescence microscopy in combination with proximity ligation assay (PLA) visualization of endogenous protein-protein interactions (red dots) within A431 cells in (A) untreated (Unt.) cells, (B) UV treated and a chase time of 30 min, (C) 200 µM $H_2O_2$ ($H_2O_2$) for 30 min, and (D) hypo-osmotic treatment (HYPO) for 10 min. Top panel: BRCA1 and cavin3; bottom panel: cavin1 and cavin3. (E) PLA signals/cell for cavin3-BRCA1 association in 50 cells/biological replicate with three independent experiments. (F) PLA time-course analysis after UV treatment and a chase time up to 360 min in 50 cells/biological replicate with three independent experiments. (G) PLA signals/cell for cavin1-cavin3 association in 50 cells/biological replicate with three independent experiments. All data was quantified from three independent experiments using a nested ANOVA. Each biological replicate is color-coded with the mean ± SEM presented as a black bar. NS: not significant; *p<0.05, **p<0.01, ***p<0.001.

The online version of this article includes the following figure supplement(s) for figure 6:

**Figure supplement 1.** Close association of cavin3 and BRCA1 in MDA-MB231 cells after stress treatment.

*Figure 6 continued on next page*

*Figure 6 continued*

**Figure supplement 2.** Proximity ligation assay (PLA) controls for cavin3 and BRCA1 PLA antibodies in A431 cells.

We propose that cavin3 can modulate BRCA1 function via multiple mechanisms: direct interaction with the RING domain of BRCA1 (*Figure 2J*), increased localization of BRCA1 to the cytosol (*Figure 4A, B*), regulation of BRCA1 protein levels (*Figure 4C, F*, *Figure 4—figure supplement 2*), modulation of proteasome-mediated protein degradation (*Figure 4G*), by facilitating the localization of components of the BRCA1-A-complex in response to UV-induced DNA damage (*Figure 8E*) and in DNA repair, as cavin3-deficient cells were sensitive to PARP inhibition, suggesting that these cells are deficient in homologous recombination DNA repair (*Figure 8F*).

We show that the ubiquitin-proteasomal degradation pathway plays a role in the coordinated protein stability of BRCA1 and cavin3 (*Figure 4G*). Previous studies have identified the RING domain region of BRCA1 as the degron sequence necessary for polyubiquitination and proteasome-mediated protein degradation, which coincides with the interaction domain of BRCA1 identified here for cavin3 (*Lu et al., 2007*). Our data further supports studies that the ubiquitin-proteasome plays an important role in regulating BRCA1 during genotoxic stress (*Lu et al., 2007*). Interaction of BRCA1 with BARD1 protein reduces proteasome-sensitive ubiquitination and stabilization of BRCA1 expression (*Choudhury et al., 2004*). BARD1 levels were downregulated in cavin3 KO cells (*Figure 1—figure supplement 1D*). Downregulation of BARD1 would be expected to impair BRCA1 function further in cavin3 KO cells as this interaction stabilizes both proteins, which then has a significant role in homologous recombination DNA repair (*Xia et al., 2003*). Further experiments are required to determine if cavin3 disrupts the interaction between BRCA1 and BARD1 and the contribution of BARD1 to the loss of BRCA1 stability and function in these cells.

In addition to its expression, BRCA1 subcellular localization is a significant contributor to its cellular functions (*Henderson, 2012*). Our findings imply that cavin3 may play a role in the cytosolic translocation of BRCA1 (*Figure 4A, B*). It is intriguing to hypothesize that BRCA1, together with cavin3, executes its tumor suppressor function by its critical role in DNA repair in the nucleus and through signaling pathways and interactions that induce the apoptotic machinery in the cytoplasm. This implies that failed repair of DNA damage in the nucleus is linked to the induction of cell death processes. The elimination of damaged cells occurs in the cytosol and that BRCA1-cavin3 may contribute to this pathway. Interestingly, cells expressing tr-BRCA1, which was identified here as the BRCA1 domain interacting with cavin3 (*Figure 2J*), have been shown to cause BRCA1 translocation to the cytosol and enhance sensitivity to UV (*Wang et al., 2010*). Ongoing investigations to test this idea may provide further insight into the role of BRCA1 nuclear-cytoplasmic shuttling and determination of cell fate (survival vs. death). Furthermore, these data also point to the potential use of BRCA1 shuttling as a novel therapeutic strategy by which manipulation of BRCA1 localization can control cellular function and sensitivity to therapy.

Cavin3 KO cells exhibited a reduction in recruitment of the BRCA1 A-complex to UV-induced DNA damage foci (*Figure 8E*). This was further correlated with a decrease in the protein levels of the components of the BRCA1 A-complex, specifically in these cells (*Figure 8D*). This is consistent with the observation that the loss of any member of the RAP80-BRCA1 complex eliminates observable BRCA1 foci formation as the BRCA1 A-complex requires all its protein components to be stable to optimally recruit BRCA1 to DSBs (*Jiang and Greenberg, 2015*). Recent studies from our laboratory have shown that γH2AX phosphorylation is compromised in cavin3 KD cells and that γH2AX forms a complex with the protein phosphatase PP1alpha, whose activity was regulated by cavin3 (*McMahon et al., 2019*). γH2AX is one of the initial factors that recruit checkpoint and DNA repair proteins to DSBs. Failure of cavin3 KO cells to phosphorylate H2AX may further compromise DNA repair mechanisms in these cells.

In addition, LFQ proteomics revealed that cavin3 KO cells upregulate many proteins involved in the protection and maintenance of the replication fork and postreplication repair, suggesting involvement of cavin3 in alternative DNA repair pathways that ultimately leads to cell survival (*Figure 1—figure supplement 1*). These pathways collectively may account for many of the characteristic features of genomic instability in familial breast and ovarian cancers, and cavin3 KO cells provide

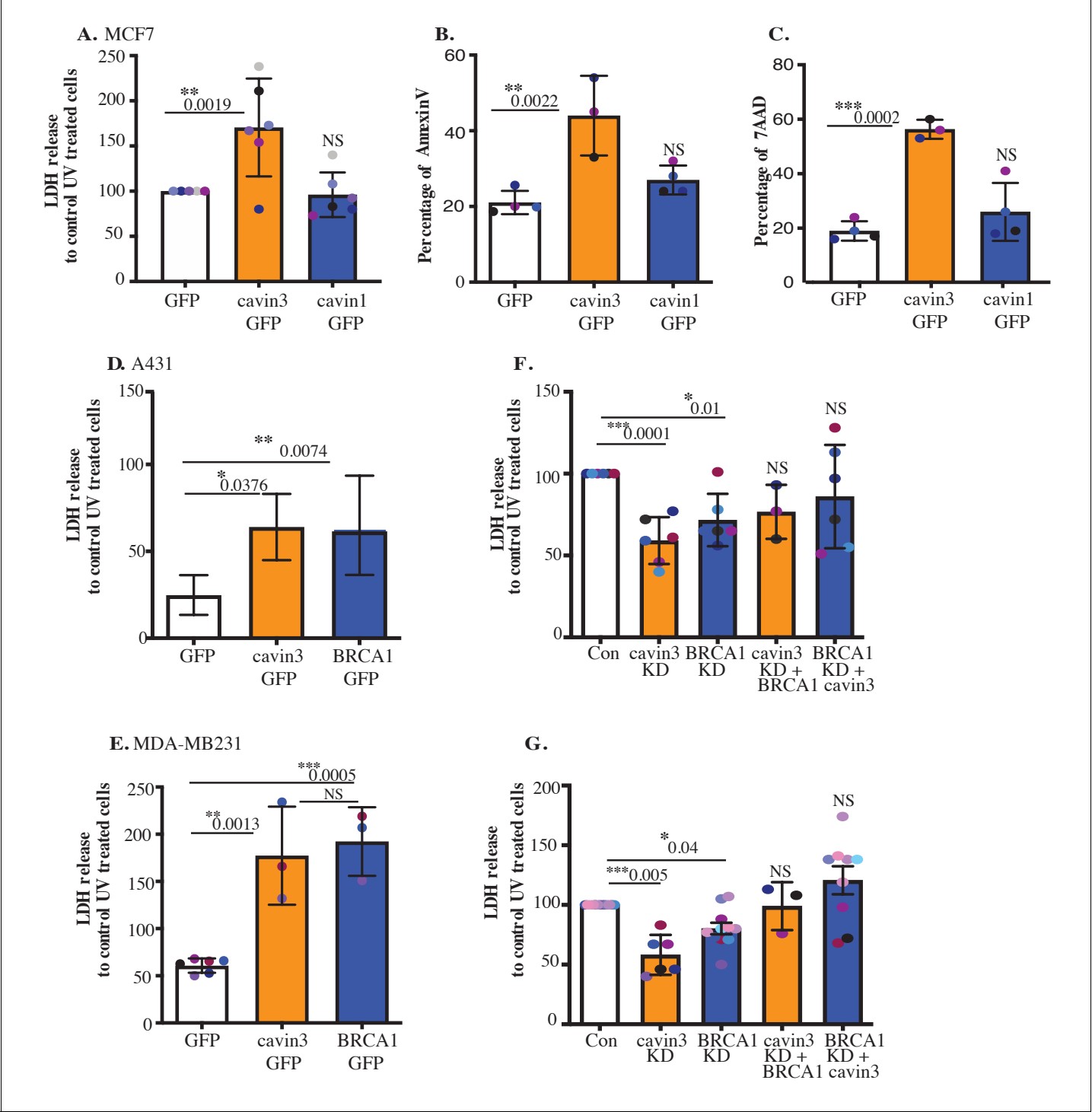

**Figure 7.** Cavin3 potentiates BRCA1 functions in apoptosis. (A) LDH release of MCF7-GFP, cavin3-GFP, and cavin1-GFP cells subjected to UV treatment and a 6 hr chase. LDH release is expressed as a percentage to control GFP cells from six independent experiments presented as mean ± SD using a one-way ANOVA and Bonferroni's multiple comparisons test. (B) Annexin V-positive cells after UV treatment and a 6 hr recovery time in MCF7 cells presented as mean ± SD using a one-way ANOVA and Bonferroni's multiple comparisons test from three independent experiments. (C) 7-AAD-positive cells after UV treatment and a 24 hr recovery time in MCF7 cells presented as mean ± SD using a one-way ANOVA and Bonferroni's multiple comparisons test from three independent experiments. (D) A431 cells and (E) MDA-MB231 cells were transfected with GFP, cavin3-GFP, or BRCA1-GFP. Results are the relative percentage of LDH release to GFP as mean ± SD using a one-way ANOVA and Bonferroni's multiple comparisons test from at least three independent experiments. (F) A431 cells and (G) MDA-MB231 cells were treated with control, cavin3, or BRCA1 specific siRNAs. Cavin3-depleted A431 and MDA-MB231 cells were transfected with BRCA1-GFP for 24 hr. BRCA1-depleted A431 and MDA-MB231 cells were transfected with

*Figure 7 continued on next page*

*Figure 7 continued*

cavin3-GFP for 24 hr. All cells were UV treated, and LDH release was measured and calculated relative to control siRNA UV-treated cells. The results represent independent experiments as mean ± SD using a one-way ANOVA and Bonferroni's multiple comparisons test from three independent experiments. Each biological replicate is color-coded. NS: not significant; *p<0.05, **p<0.01, ***p<0.001.

The online version of this article includes the following figure supplement(s) for figure 7:

**Figure supplement 1.** Validation of LDH release in MCF7, A431, and MDA-MB231 cells.

**Figure supplement 2.** Cavin3 KO cells exhibit resistance to stressors that allow BRCA1 interaction.

an alternative model cell line for further investigation (see *Supplementary file 3* for further analysis of cavin3-dependent pathways).

Recent clinical evidence has shown that mutations in BRCA1 do not entirely account for the treatment benefits seen with PARP inhibitors (*O'Shaughnessy et al., 2011*; *Javle and Curtin, 2011*; *Pilié et al., 2019*). Loss of cavin3 expression has been observed in many human malignancies (*Carén et al., 2011*; *Kim et al., 2014*; *Lee et al., 2008*; *Lee et al., 2011*; *Martinez et al., 2009*; *Tong et al., 2010*; *Xu et al., 2001*; *Zöchbauer-Müller et al., 2005*). Several studies have shown that low expression of cavin3 promotes cisplatin resistance and oxaliplatin resistance in lung and colorectal cancers, respectively (*Fu et al., 2020*; *Moutinho et al., 2014*). This is in contrast to BRCA1-deficient cells that are sensitive to these platinum drugs (*Mylavarapu et al., 2018*). These findings suggest that knowing the status of cavin3 in tumors in addition to BRCA1 may be used to better stratify patients in predicting drug sensitivity, that is, PARP inhibitors versus platinum drugs in the clinic. These findings also suggest that cavin3 KO cells may provide a unique platform to understand platinum drug resistance in the absence of BRCA1 expression. This may involve alterations in non-homologous end-joining repair, replication fork protection, upregulation of cellular drug efflux pumps, and alterations to the tumor microenvironment that can now be explored in these cells.

Previous studies have shown that cavin3 knockout mice are not cancer-prone (*Hernandez et al., 2013*). This raises the question as to how cavin3 may act as a tumor suppressor. Cavin3 inactivation may contribute to tumor progression by reducing cellular sensitivity to stressors as shown here as well as in previous published studies contributing to overall cell survival (*Lee et al., 2011*). Cavin3 mRNA is increased in response to numerous stresses, suggesting regulation by stress signaling and cellular damage (*Lee et al., 2011*). This may involve p53 as cavin3 increases the stability of p53 and its target gene expression and loss or reduction in tumor cells lessen p53 response to stresses, which contribute to malignant tumor progression (*Lee et al., 2011*). Here, we have shown that cavin3 also interacts with BRCA1 where the two proteins work together to regulate DNA repair or, in extreme conditions, trigger apoptosis. Collectively our studies suggest that loss of cavin3 function might provide tumor cells' survival and growth advantages by attenuating the apoptotic sensitivity to various stresses and hindering DNA repair under chronic stress conditions.

Loss of cavin3 expression is more prevalent in late-stage/high-grade cancers than in early-stage/low-grade cancers (*An et al., 2020*; *Carén et al., 2011*; *Lee et al., 2008*; *Wikman et al., 2012*). Cavin3 expression is lost due to promoter methylation in numerous cancer types (*Lee et al., 2008*; *Lee et al., 2011*; *Martinez et al., 2009*; *Tong et al., 2010*; *Xu et al., 2001*; *Zöchbauer-Müller et al., 2005*). Silencing of a DNA repair gene such as cavin3 by hypermethylation may be a very early step in the progression to cancer (*Jin and Robertson, 2013*). Such silencing is proposed to act similarly to a germline mutation in a DNA repair gene and predisposes these cells to cancer. This may occur through deficiency in DNA repair. This would allow for accumulation of DNA damage causing increased errors during DNA synthesis, leading to mutations that can give rise to cancer. This may further contribute to the tumor suppressor functions of cavin3.

Finally, the example of cavin3 leads us to propose a general model for cell stress sensing mediated by cavins when they are released from caveolae to interact with intracellular targets. Rigorous control of such a pathway would require that cytosolic levels of cavins be kept low under steady-state conditions. Recent work shows that this can be achieved by ubiquitination of a conserved phosphoinositide-binding patch on cavins that is only exposed when cavins are released from caveolae (*Tillu et al., 2015*). In the absence of stabilizing interactions, the released cavin protein will undergo proteasomal degradation, but, as shown here, interaction with BRCA1 stabilizes cavin3, preventing degradation. We propose that the interaction of cavin3 with BRCA1 in response to short-term stress

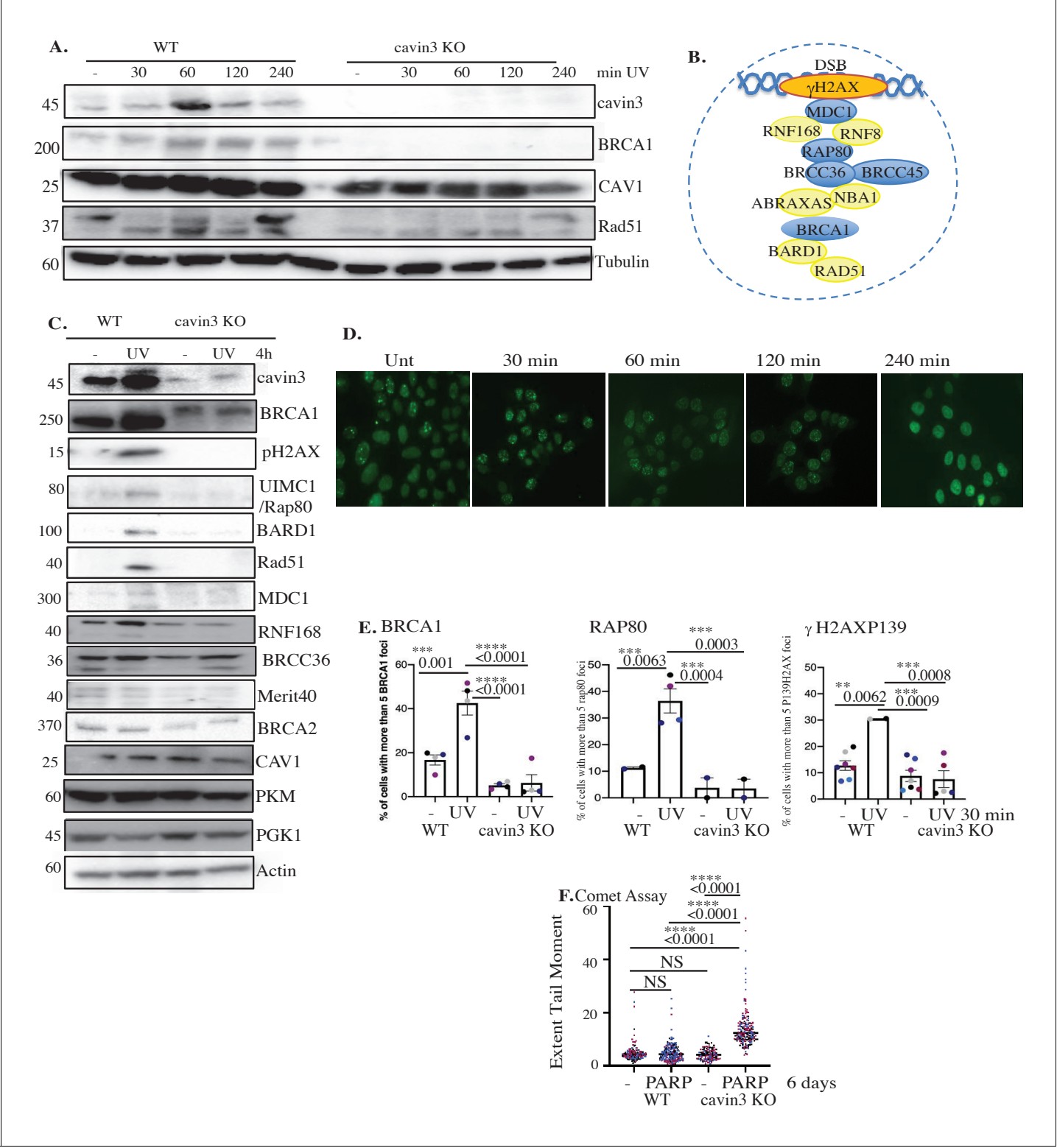

**Figure 8.** Cavin3-deficient HeLa cells exhibit abolishment of DNA repair. (**A**) Representative western blot analysis of WT and cavin3 KO cells UV time course for cavin3, BRCA1, CAV1, Rad51, and Tubulin. (**B**) Protein components of the BRCA1 A-complex. Blue-colored circles: proteins downregulated in the label-free quantitative (LFQ) proteomics; yellow-colored circles: proteins not detected in the LFQ proteomics of cavin3 KO cells. (**C**) Representative western blot analysis of cavin3, BRCA1, pH2AX, UIM1C/Rap80, BARD1, Rad51, MDC1, RNF168, BRCC36, Merit40, BRCA2, CAV, PKM, PGK1, and Actin in WT and cavin3 KO HeLa cells untreated (-) or UV treated (UV) followed by a 4 hr chase. Quantitation of protein levels from three independent

*Figure 8 continued on next page*

*Figure 8 continued*

experiments is presented in *Figure 8—figure supplement 1*. (D) Representative immunofluorescence images of BRCA1 foci after UV treatment in WT HeLa cells. (E) Percentage of cells with more than five BRCA1 foci, Rap80 foci, and γH2AX foci in WT and cavin3 KO cells following UV treatment and a 30 min chase. The results are presented as mean ± SD using a one-way ANOVA and Bonferroni's multiple comparisons test from three independent experiments. (F) WT and cavin3 KO cells untreated or treated with the PARP inhibitor (AZD2461, PARPi) 5 nM for 6 days were subjected to comet assays. The results are presented as the mean ± SEM using a one-way ANOVA and Bonferroni's multiple comparisons test from three independent experiments. Each biological replicate is color-coded. Extent Tail Moment was calculated as described in Materials and methods. NS: not significant; **p<0.01, ***p<0.001; ****p<0.0001.

The online version of this article includes the following source data and figure supplement(s) for figure 8:

**Source data 1.** Raw western data for HeLa WT and cavin3 KO cells time course after UV treatment with molecular weight markers for *Figure 8A*.

**Source data 2.** Raw western data for HeLa WT and cavin3 KO cells untreated or UV treatment for 4 hr with molecular weight markers for *Figure 8C*.

**Figure supplement 1.** Quantitation of BRCA1-A-complex proteins in WT and cavin3 KO cells.

**Figure supplement 2.** Cavin3 KO cells are sensitive to PARP inhibition, and 53BP1 loss causes PARP inhibitor reversion.

**Figure supplement 2—source data 1.** Raw western data for HeLa WT and cavin3 KO cells depleted of FANCD2, PARP1, CHD3, and 53BP1 with molecular weight markers for *Figure 8—figure supplement 2*.

can facilitate DNA repair. With a prolonged stress, this can trigger apoptosis as a protective mechanism. This forms a novel signaling pathway to protect cells against many cellular stresses and represents a new paradigm in cellular signaling that can explain the evolutionary conservation of caveolae and their involvement in multiple signal transduction pathways.

In view of the loss of cavin3 in numerous cancers (*Carén et al., 2011*; *Kim et al., 2014*; *Lee et al., 2008*; *Lee et al., 2011*; *Martinez et al., 2009*; *Tong et al., 2010*; *Xu et al., 2001*; *Zöchbauer-Müller et al., 2005*) and the crucial role of BRCA1 as a tumor suppressor (*King and Marks, 2003*; *Miki et al., 1994*; *Venkitaraman, 2002*), these studies describing a new functional partner for BRCA1 suggest that cavin3 should be considered in future cancer diagnostic and therapeutic strategies.

## Materials and methods

### Reagents

Dulbecco's modified Eagle's medium (DMEM, Cat# 10313-021), Z150 L-glutamine 100× (Cat# 25030-081), and Trypsin-EDTA (0.05%) phenol red (Cat# 25300062) were from Gibco by Life Technologies, Australia. SERANA fetal bovine serum (FBS) (Cat# FBS-AU-015, batch no. 18030416) was from Fisher Biotechnology, Australia. cOmplete, mini EDTA-free protease inhibitor cocktail (Cat# 11836170001), PhosSTOP Phosphatase Inhibitors (Cat# 4906837001), hydrogen peroxide 30% (w/w) solution (Cat# H1009), AZD2461 (Cat# SML 1858), and MG132 (Z-Leu-Leu-Leu-al, Cat# C2211) were from Sigma-Aldrich.

### Antibodies

The following antibodies were used: rabbit anti-53BP1 (Cat# GTX 112864, GeneTex, WB 1:1000), rabbit anti-ACCA antibody (Cell Signaling, Cat# 3662, RRID:AB_2219400, WB 1:5000), mouse anti-Actin antibody (Millipore, Cat# MAB1501, RRID:AB_2223041, WB 1:5000), rabbit anti-ACLY antibody (Sigma-Aldrich, Cat# HPA028758, RRID:AB_10603575, WB 1:2000), mouse anti-Aurora kinase antibody (BD Biosciences, Cat# 611082, RRID:AB_2227708, PLA 1:100), mouse-anti-BARD1 E-11 antibody (Santa Cruz, Cat# sc-74559, RRID:AB_2061237, WB 1:500), rabbit anti-BRCA1 20 antibody (Santa Cruz, Cat# sc-642, RRID:AB_630944, WB 1:500, IF 1:100, PLA 1:100), mouse anti-BRCA1 MS110 antibody (Abcam, Cat# ab16780, RRID:AB_2259338, WB 1:1000, IF 1:100, PLA 1:100), mouse-anti-BRCA1 D-9 antibody (Santa Cruz, Cat# sc-6954, RRID:AB_626761, IF 1:50), rabbit-anti-BRCA1 antibody (Millipore, Cat# 07-434, RRID:AB_2275035, WB 1:2000), rabbit-anti-BRCA1 antibody (Proteintech, Cat# 22363-1-AP, RRID:AB_2879090, WB 1:1000), rabbit anti-BRCA2 antibody (BioVision, Cat# 3675-30T, RRID:AB_2067764, WB 1:2000), rabbit anti-BRCC36 antibody (ProScience, Cat# 4311, WB 1:1000), rabbit anti-BRCC45 antibody (GeneTex, Cat# GTX105364, RRID:AB_1949757, WB 1:2000), mouse anti-Caldesmon antibody (BD Biosciences, Cat# 610660, WB 1:3000), mouse anti-alpha catenin antibody (Cell Signaling, Cat# 2131, WB 1:3000), mouse anti-

gamma catenin antibody (Cell Signaling, Cat# 2309, WB 1:3000), rabbit anti-CAV1 antibody (BD Biosciences, Cat# 610060, WB 1:5000), mouse anti-cavin1 antibody (Abmart, China, 1:100 PLA), and rabbit anti-cavin1 antibody were raised as described previously and used for immunofluorescence (*Bastiani et al., 2009*), rabbit anti-cavin1 antibody (Sigma-Aldrich, Cat# AV36965, RRID:AB 1855947, WB 1:2000), mouse anti-cavin3 antibody (Novus, Cat# H00112464-MO4, PLA 1:200), rabbit anti-cavin3 antibody (Proteintech, Millennium Sciences, Pty, Ltd, Cat# 16250-1-AP, RRID:AB_ 2171897, WB 1:2000, IF 1:300, PLA 1:200), rabbit anti-CHD3 antibody (GeneTex, Sapphire Bioscience, Cat# GTX131779, RRID:AB_2886520, WB 1:500), rabbit anti-DDX21 antibody (Novus, Cat# NBP1-88310, RRID:AB_11027665, WB 1:2000), rabbit anti-EGFR Clone LA22 antibody (Millipore, Cat# 05-104, RRID:AB_11210086, WB 1:4000), mouse-anti-FANCD2 antibody (GeneTex, Cat# GTX116037, RRID:AB2036898, WB 1:500), mouse anti-Flotillin Clone 18 antibody (BD Biosciences, Cat# 610821, RRID:AB_398140, PLA 1:100), mouse anti-GFP antibody (Roche, Cat# 11814460001, RRID:AB_390913, WB 1:4000, PLA 1:300), rabbit anti-Histone H2A.X-Chip Grade (Abcam, Cat# ab20669, RRID:AB_445689, WB 1:1000), rabbit phospho-Histone H2A.X (Ser 139) (20E3) antibody (Cell Signaling Technology, Cat# 9718, RRID:AB_2118009, IF 1:500), rabbit phospho-Histone H2A.X CHIP Grade antibody (Abcam, Cat# ab2893, RRID:AB_303388, WB: 1:3000), rabbit anti-HLTF antibody (Proteintech, Cat# 14286-1-AP, WB 1:2000), rabbit anti-MDC1 antibody (Novus, Cat# 10056657SS, RRID:AB_838567, WB 1:100), sheep anti-Merit40 antibody (R&D Systems, Cat# AF6604SP, RRID:AB_10717577, WB 1:500), rabbit anti-PARP1 antibody (GeneTex, Cat# GTX112864, RRID:AB_11173565, WB 1:1000), mouse anti-PCNA antibody (Millipore, Cat# NA03T, RRID:AB_ 2160357, PLA 1:100), rabbit anti-PGK1 antibody (GeneTex, Cat# GTX107614, RRID:AB_2037666, WB 1:3000), rabbit anti-PKM antibody (GeneTex, Cat# GTX107977, RRID:AB_1951264, WB 1:3000), mouse anti-Rad51 antibody (Novus, Cat# 100-184, RRID:AB_350083, WB 1:1000), rabbit anti-RAP80 D1T6Q antibody (Cell Signaling Technology, Cat# 14466, RRID:AB_2798487, WB 1:1000, IF 1:100), rabbit anti-RNF168 antibody (GeneTex, Cat# GTX118147, RRID:AB_11169617, WB 1:1000), and mouse anti-Tubulin DM1A antibody (Abcam, Cat# ab7291, RRID:AB_2241126, WB 1:4000).

Secondary antibodies for immunofluorescence were Alexa Fluor 488 Goat anti-Rabbit IgG (H + L) (Thermo Fisher Scientific, Cat# A-11034, RRID:AB_141637, IF 1:500), Alexa Fluor 546 Goat anti-Mouse IgG (H + L) (Thermo Fisher Scientific, Cat# A-11030, RRID:AB_2534089, IF 1:500), Alexa Fluor 594 Donkey anti-Rabbit IgG (H + L) (Thermo Fisher Scientific, Cat# A-21207, RRID:AB_141637, IF 1:500), and Alexa Fluor 594 Goat anti-Mouse IgG (H + L) (Thermo Fisher Scientific, Cat# A-21203, RRID:AB_141633, IF 1:500). Secondary antibodies for western blotting were Goat anti-Mouse IgG (H + L) cross adsorbed secondary antibody, HRP (Thermo Fisher Scientific, Cat# G-21040, RRID:AB_ 2536527, WB 1:5000), Goat anti-Mouse IgG (H + L) cross adsorbed secondary antibody, HRP (Thermo Fisher Scientific, Cat# G-21234, RRID:AB_2536527, WB 1:5000), and Rabbit anti-Sheep IgG (H + L) (Abcam, Cat# ab97130, RRID:AB_2536530, WB 1:2000).

## Cell culture

MCF7 cells, a human adenocarcinoma cell line with a low invasive phenotype (ATTC HBT-22, RRID: CVCL_0031), were subjected to STR profiling (QIMR Berghofer Cancer Research Institute). MDA-MB231 cells (ATCC HTB-26, RRID:CVCL_0062), a human adenocarcinoma cell line, and A431 cells (ATCC CRL-1555, RRID:CVCL_0037), HeLa cells (ATCC CRM-CCL2, RRID:CVCL_0030), and HeLa KO for cavin3 were cultured in DMEM supplemented with 10% (vol/vol) FBS, 100 units/ml penicillin, and 100 µg/ml streptomycin. All cell lines were routinely tested for mycoplasma. MCF7 cells were seeded at $1 \times 10^6$ cells and were transfected with 5 µg pEGFP DNA, pEGFP-cavin1, pEGFP-cavin2, pEGFP-cavin3, or pEGFP-CAV1 DNA using Lipofectamine 2000 (Invitrogen) according to the manufacturer's instructions. G418 (Sigma-Aldrich, Cat# 472788001) was used as a selection drug at 500 µg/ml.

## Generation of CRISPR cavin3 knockout cell lines

The HeLa cavin3 KO cell line was generated as follows according to the protocol published previously (*Stroud et al., 2016*). Targeting was to the first exon at the second in-frame ATG about one-third through the exon as this was easy for targeting.

### Zifit input (in-frame ATGs, target site)

CAVIN3: GGGGCCTGTGCCCGAGGCGCCGGCGGGGGGTCCCGTGCACGCCGTGACGG
TGGTGACCCTGCTGGAGAAGCTGGCCTCCATGCTGGGAGACTCTGCGGGAGCGGCAGG-
GAGGCCTGGCTCGAAGGCAGGGAGGCCTGGCAGGGTCCGTGCGCCGCATCCA-
GAGCGGCCTGGGCGCTCTGAGTCGCAGCCACG

### Zifit output

TALENs

Clonal cells were isolated by dilution into a 96-well plate. Total extract of single clones were prepared and analyzed by western blotting using rabbit polyclonal anti-cavin3 antibody (Millennium Science). Total deletion of cavin3 was verified by PCR and western analysis (*Figure 1—figure supplement 1A, B*).

### Immunofluorescence

In brief, MCF7, MDA-MB231, and A431 cells seeded onto glass coverslips at 70% confluence were washed once in PBS and were then fixed in 4% (vol/vol) PFA in PBS for 20 min at room temperature (RT). Coverslips were washed three times in excess PBS and were permeabilized in 0.1% (vol/vol) Triton X-100 in PBS for 7 min and blocked in 1% (vol/vol) bovine serum albumin (BSA) (Sigma-Aldrich) in PBS for 30 min at RT. The primary antibodies were diluted in 1% (vol/vol) BSA in PBS and incubated for 1 hr at RT. Secondary antibodies (Molecular Probes) were diluted in 1% (vol/vol) BSA in PBS and incubated for 1 hr at RT. Washes were performed in PBS. Coverslips were rinsed in distilled water and mounted in Mowiol (Mowiol 488, Hoechst AG) in 0.2 M Tris-HCl, pH 8.5. The images were taken on a laser-scanning microscope (LSM 510 META, Carl Zeiss, Inc) using a 63× oil lens, NA 1.4. Adjustments of brightness and contrast were applied using ImageJ software (NIH). The LUT of images for PLA were inverted for better visualization of PLA dots in cells.

### Foci immunofluorescence

HeLa WT and cavin3 KO cells were pre-permeabilized with CSK buffer (10 mM HEPES, 100 mM NaCl, 300 mM sucrose, 3 mM $MgCl_2$, 0.7% Triton X-100) for 5 min and were then fixed with 4% PFA/PBS for 15 min, permeabilized with 0.5% Triton X-100 solution for 15 min followed by blocking for 1 hr RT. Cells were then immunostained with primary antibodies against mouse BRCA1 alone (Santa Cruz, Cat# sc-6954, RRID:AB_626761, IF 1:50), Rap80 alone (Cell Signaling Technology, Cat# 14466, RRID:AB_2798487, IF 1: 50), γH2AX alone (Abcam, Cat# 20669, RRID:AB_445689, IF 1:100), and the appropriate Alexa Fluor 488 Goat anti-Rabbit IgG (H + L) (Thermo Fisher Scientific, Cat# A-11034, IF 1:500) conjugated secondary antibodies. Images were taken with a Zeiss microscope. Quantification of the percent of cells was based on foci formation (more than 5 foci/nucleus) was determined from more than 500 cells/experimental condition from 2 to 3 independent experiments using an automated plugin for ImageJ.

### Proximity ligation assay

Detection of an interaction between BRCA1 and the cavin or CAV1 proteins was assessed using the Duolink II Detection Kit (Sigma-Aldrich) according to the manufacturer's specifications. The Duolink In situ PLA Probe Anti-Rabbit MINUS (Sigma-Aldrich, DUO92005, RRID:AB_2810942) and Duolink In situ PLA Probe anti-Mouse PLUS (Sigma-Aldrich, DUO92001, RRID:AB_281039) and Duolink In situ detection reagents Orange (DUO 92007) were used in all PLA experiments. The primary antibodies used were mouse monoclonal GFP (1:500) and rabbit polyclonal BRCA1 (1:200), rabbit cavin3 (1:200) and mouse PCNA (1:100), rabbit cavin3 (1:200) and mouse Aurora Kinase (1:100), rabbit cavin3 (1:200) and Flotillin (1:100), and cavin3 (1:200) and mouse cavin 1 (1:100). The signal was visualized as a distinct fluorescent spot and was captured on an Olympus BX-51 upright Fluorescence Microscope. The number of PLA signals in a cell was quantified in ImageJ using a Maximum Entropy Threshold and Particle Analysis where 50 cells in each treatment group were analyzed from at least three independent experiments.

## SDS-PAGE and western blot analysis

For SDS-PAGE, cells were harvested, rinsed in PBS, and were lysed in lysis buffer containing 50 mM Tris pH 7.5, 150 mM NaCl, 5 mM EDTA pH 8.0, 1% Triton X-100 with protease and phosphatase inhibitors. Lysates were collected by scraping and cleared by centrifugation at 4°C. The protein content of all extracts was determined using the Pierce BCA Protein Assay Kit (Cat# 23225, Thermo Fisher Scientific) using BSA as the standard. 30 µg of cellular protein were resolved by 10% SDS-PAGE and were transferred to Immobilin P 0.45 mm PVDF membrane (Merck). Bound IgG was visualized with horseradish peroxidase-conjugated secondary antibodies and the Clarity Western ECL Substrate (Cat# 1705061, Bio-Rad, Gladesville, New South Wales, Australia).

## Stress experiments

A431 or MDA-MB231 cells were plated on coverslips at 70% confluency. Cells were either left untreated or were treated with 200 µM $H_2O_2$ for 30 min, 90% hypo-osmotic media for 10 min, or UV treatment for 2 min without media with a UV germicidal light source (UV-C 254 nm) and allowed to recover for 30 min in complete cell culture medium as previously described in *McMahon et al., 2019*. All cells were fixed and processed for cavin3 and BRCA1 or cavin3 and cavin1 using the PLA as described.

## PrestoBlue cell viability assays

HeLa WT and cavin3 KO cells were counted using a hemocytometer and seeded into 96-well plate at 1000 cells/well (eight wells for each treatment) in 90 ml medium per well. Cells were either left untreated or were treated with 90% hypo-osmotic media (90% water in DMEM), UV treatment for 2 min without media with a UV germicidal light source (UV-C 254 nm) or 200 µM $H_2O_2$. After stress addition, 10 ml of PrestoBlue Viability Reagent (10×) (Absorbance wavelength: 600 nm) (Thermo Fisher Scientific) was added to cells. The PrestoBlue reagent was incubated constantly in wells over a time course from 2 hr to 24 hr. Control wells containing only cell culture media (no cells) were included in triplicate on each plate for background fluorescence calculations. Plates were returned to a 37°C incubator. Both absorbance values at 570 nm and 600 nm were measured for each plate in a TECAN Infinite 200 Pro reader (Millennium Science), where 570 nm was used as the experimental wavelength and 600 nm as normalization wavelength.

For PARP inhibitor experiments, cells were either left untreated or were treated with PARP inhibitor (AZD2461 5 nM) for 6 days after which 10 µl of PrestoBlue Viability Reagent (10×) (absorbance wavelength: 600 nm) (Thermo Fisher Scientific) was added to cells. Control wells containing only cell culture media (no cells) were included in triplicate on each plate for background fluorescence calculations. Plates were returned to a 37°C incubator. Both absorbance values at 570 nm and 600 nm were measured for each plate in a TECAN Infinite 200 Pro reader, where 570 nm was used as the experimental wavelength and 600 nm as normalization wavelength.

Raw data was processed to evaluate the percent reduction of PrestoBlue reagent for each well by using the following equation referring to the manufacturer's protocol:

$$\%\,\mathrm{Reduction\,in\,Prestoblue} = \frac{(117216 \times A1) - (80586 \times A2)}{(155677 \times A1) - (14652 \times A2)} \times 100$$

where **A1** is the absorbance of test wells at 570 nm, **A2** is the absorbance of test wells at 600 nm, **N1** is the absorbance of media-only wells at 570 nm, and **N2** is the absorbance of media-only wells at 600 nm.

## RNA interference

Human cavin3 Stealth siRNAs (set of three – HSS174185, 150811, 150809) and Human BRCA1 Stealth siRNAs (set of three – HSS101089, 186096, 186097) were purchased from Life Technologies Australia Pty Ltd. Two siRNA oligonucleotides to cavin3 or BRCA1 were found to reduce protein levels (oligo 1 and oligo 2) and were transfected into cells at 24 hr and 48 hr after plating using Lipofectamine 2000 reagent (Invitrogen) with a ratio of 6 µl Lipofectamine to 150 pmol siRNA. Cells were split and harvested after 72–96 hr for further analysis.

## CRISPR-Cas9-mediated gene knockouts

WT and cavin3 KO cells lacking CHD3, FANCD2, PARP1, and TP53BP1 were generated using the Alt-R CRISPR-Cas9 system (Integrated DNA Technologies). The following predesigned Alt-R CRISPR-Cas9 gRNAs were used:

> Hs.Cas9.CHD3.1.AA, strand sequence GACCGGGTCGGAAACGAAGA
> Hs.Cas9.FANCD2.1.AA, strand sequence AGTTGACTGACAATGAGTCG
> Hs.Cas9.PARP1.1.AA, strand sequence GAGTCGAGTACGCCAAGAGC
> Hs.Cas9.TP53BP1.1.AA strand sequence AACGAGGAGACGGTAATAGT

Each RNA oligo (Alt-R CRISPR Cas9 cRNA, tracrRNA) was resuspended in Nuclease-Free IDTE Buffer. The crRNA and tracrRNA were mixed in equimolar concentrations, heated at 95°C for 5 min, followed by cooling to RT. To produce the RNP complex for each well of a 96-well plate, the following was combined: 1.5 µl of 1 µM Guide RNA oligos, 1.5 µl of 1 µM diluted Cas9 enzyme with 0.6 µl of Cas9 PLUS Reagent from Lipofectamine CRISPRMAX kit and 21.4 µl of Opti-MEM Media followed by incubation at RT for 5 min to assemble the RNP complexes. The RNP was further mixed with 1.2 µl of CRISPRMAX transfection reagent in Opti-MEM for a further 20 min to form the transfection complexes. This was then added to 40,000 HeLa WT or cavin3 KO cells/ml that were seeded in a well of a 96-well tissue culture plate. The plates containing the transfection complexes and cells were returned to a tissue culture incubator for 72 hr. These cells were then subjected to single-cell plating for clonal selection. Loss of each of the proteins was verified by western blot analysis of cell lysates using the following antibodies: CHD3 (GeneTex, Cat# GTX131779, RRID:AB_2886520, WB: 1:500), FANCD2 (GeneTex, Cat# GTX116037, RRID:AB_2036898, WB: 1:500), PARP1 (GeneTex, Cat# GTX112864, RRID:AB_11173565, WB: 1:1000), and 53BP1 (GeneTex, Cat# GTX70310, WB 1:1000).

## Clonogenic survival assays

WT HeLa and cavin3 KO cells were seeded at low density (500 cells/well) in six-well plates, left untreated or treated with 5 nM concentrations of PARP (AZD2461), and were allowed to form colonies for 6 days. Colonies were fixed and stained with 0.5% crystal violet/20% ethanol and counted. Results were normalized to plating efficiencies where the

> Plating efficiency (PE) = no. of colonies formed/no. cells seeded $\times$ 100% and
> Survival fraction (SF) = no. of colonies formed after PARP treatment/no. cells seeded $\times$ PE $\times$ 100%

## Comet assay

Comet microscopes slides were prepared the day before the assay by melting low melting point 0.5% agarose in a microwave until the agarose was completely molten. Thoroughly cleaned glass microscope slides were layered with the agarose. Slides were left on a flat surface to air-dry overnight where a transparent agarose film formed after drying. Coated slides were placed at 37°C before use.

HeLa WT and cavin3 KO cells either left untreated or treated with UV (2 min) and a 4 hr recovery time were trypsinized, and cells were suspended at $2 \times 10^5$ cell/mL in $1\times$ PBS. The cell samples were prepared immediately before starting the assay, and all samples were handled in a dimmed environment to prevent DNA damage from light. The cell suspension was mixed with 0.5% molten low melting point agarose (at 37°C) at a ratio 1:10 (v/v). Cells were mixed gently by pipetting up and down and immediately added on top of the agarose layer on the glass slides. The side of the pipette tip was used to spread the agarose/cell mixture to ensure the formation of a thin layer. Slides were then placed at 4°C in the dark for 30 min. Slides were then carefully immersed in lysis buffer (2.5M NaCl, 0.1 M EDTA pH 8.0, 10 mM Tris, where the pH was adjusted to 10.0 with NaOH pellets and chilled before use) at 4°C in the dark for 1 hr. Slides were then immersed in alkaline solution at 4°C in the dark for 30 min. Slides were gently removed from the alkaline solution and then gently immersed in chilled $1\times$ TBE solution for 10 min in the dark. Prechilled TBE buffer was added in the electrophoresis slide tray, and the slides were placed inside for electrophoresis. The power supply was set to voltage of 1 V/cm (the length between electrodes) and run for 15 min at 4°C.

Excess buffer was removed from the slides, which were then immersed in three changes of chilled dH$_2$O for 2 min. Slides were then gently immersed in chilled 70% ethanol for 5 min at RT in the dark. Slides were then allowed to dry. 50 µl green fluorescent nucleic acid staining solution (Vista green) was then added onto each slide and was stained for 15 min at RT in the dark. The visualization and quantification of DNA breaks was based on epifluorescence microscopy. Randomly captured images from the stained comet slides were from a fluorescence microscope with a 10× objective lens. The DNA damage was quantified by measuring the displacement between the genetic material of the nucleus ('comet head') and the resulting 'tail' using ImageJ software. At least 50–100 cells were analyzed per sample from three independent experiments. The following equations were used in the analysis:

Tail DNA% = 100 × Tail DNA Intensity/Cell DNA Intensity,
Extent Tail Moment = Tail DNA% × Length of Tail where the Tail Moment Length is measured from the center of the head to the center of the tail.

## Apoptosis assay

Equal numbers of subconfluent MCF7 cells expressing GFP alone, cavin1-GFP, cavin2-GFP, cavin3-GFP, and CAV1-GFP were seeded on coverslips. Twenty-four hours later, cells were subject to UV-C exposure for 2 min without media. Complete medium lacking phenol red was added to the cells that were left at 37°C to recover. LDH release assay was measured in triplicate samples from 50 µl of conditioned media expressing cells using the Cytotoxicity Detection Kit[PLUS] (LDH) from Sigma-Aldrich according to the manufacturer's instructions. Post-nuclear supernatant from UV exposure cells was also prepared and subjected to western blot analysis with antibodies to BRCA1 (WB 1:500), GFP (WB 1:3000), and Tubulin (WB 1:5000). For knockdown experiments of cavin3 and BRCA1, after 72 hr of knockdown, cells were left untreated or were further transfected with BRCA1-GFP or cavin3-GFP overnight, respectively, and were then subjected to UV exposure 2 min and a recovery time of 6 hr. LDH release was then measured from the cell supernatant in triplicate as indicated in the respective figure legends.

## Single-molecule spectroscopy

Single-molecule spectroscopy was performed. *Leishmania* cell-free lysates were prepared according to *Kovtun et al., 2011*; *McMahon et al., 2019*; *Mureev et al., 2009*. Where indicated, MDA-MB231 or MCF7 cells were transiently cotransfected with BRCA1-GFP and mCherry alone as the control, cavin1-Cherry, cavin2-Cherry, cavin3-Cherry, or CAV1-Cherry constructs. A PNS fraction from the MDA-MB231 and MCF7 cells was prepared in 1× PBS with protease and phosphatase inhibitors for analysis. Single-molecule coincidence measurements were performed using pairs of tagged proteins to ascertain their interaction. One protein of the pair was tagged with GFP, and the other with mCherry, and both were diluted to single-molecule concentrations (~1 nM). Two lasers, with wavelengths of 488 nm and 561 nm (to excite GFP and mCherry, respectively), were focused to a confocal volume using a 40×/1.2 NA water immersion objective. The fluorescence signal from the fluorophores was collected and separated into two channels with a 565 nm dichroic. The resulting GFP and mCherry signals were measured after passing through a 525/20 nm bandpass and 580 nm long-pass filter, respectively. The signal from both channels was recorded simultaneously with a time resolution of 1 ms, and the threshold for positive events was set at 50 photons/ms. The coincidence ratio (C) for each event was calculated as C = mCherry/(GFP + mCherry), after subtracting a 6% leakage of the GFP signal into the mCherry channel. Coincident events corresponded to ~0.25 < C < 0.75. After normalizing for the total number of events (>1000 in all cases), a histogram of the C values for the protein pair was fitted with 3 Gaussians, corresponding to signals from solely GFP (green), coincidence (yellow), and solely mCherry (red).

## Quantitative mass spectrometry using HeLa WT and cavin3 KO cells

Samples were prepared for mass spectrometry and analyzed as previously described (*Stroud et al., 2016*). Briefly, cells were lysed in 1% (w/v) sodium deoxycholate, 100 mM Tris-HCl (pH 8.1), Tris(2-carboxyethy)phosphine (TCEP), 20 mM chloroacetamide, and incubated at 99 °C for 10 min. Reduced and alkylated proteins were digested into peptides using trypsin by incubation at 37 °C overnight, according to the manufacturer's instructions (Promega). Detergent was removed from the

peptides using SDB-RPS stage tips as described (*Stroud et al., 2016*). Peptides were reconstituted in 0.1%% trifluoroacetic acid (TFA), 2% ACN, and analyzed by online nano-HPLC/electrospray ionization-MS/MS on a Q Exactive Plus connected to an Ultimate 3000 HPLC (Thermo Fisher Scientific). Peptides were loaded onto a trap column (Acclaim C18 PepMap nano Trap $\times$ 2 cm, 100 µm I.D, 5 µm particle size, and 300 Å pore size; Thermo Fisher Scientific) at 15 µl/min for 3 min before switching the pre-column in line with the analytical column (Acclaim RSLC C18 PepMap Acclaim RSLC nanocolumn 75 µm $\times$ 50 cm, PepMap100 C18, 3 µm particle size 100 Å pore size; Thermo Fisher Scientific). The separation of peptides was performed at 250 nl/min using a nonlinear ACN gradient of buffer A (0.1% FA, 2% ACN) and buffer B (0.1% FA, 80% ACN), starting at 2.5% buffer B to 35.4% followed by ramp to 99% over 278 min. Data were collected in positive mode using Data Dependent Acquisition using m/z 375–1575 as MS scan range, HCD for MS/MS of the 12 most intense ions with $z \geq 2$. Other instrument parameters were MS1 scan at 70,000 resolution (at 200 m/z), MS maximum injection time 54 ms, AGC target 3E6, normalized collision energy was at 27% energy, isolation window of 1.8 Da, MS/MS resolution 17,500, MS/MS AGC target of 2E5, MS/MS maximum injection time 54 ms, minimum intensity was set at 2E3, and dynamic exclusion was set to 15 s. Thermo raw files were processed using the MaxQuant platform (*Tyanova et al., 2016*) version 1.6.5.0 using default settings for a label-free experiment with the following changes. The search database was the UniProt human database containing reviewed canonical sequences (June 2019) and a database containing common contaminants. 'Match between runs' was enabled with default settings. Maxquant output (proteinGroups.txt) was processed using Perseus (*Tyanova et al., 2016*) version 1.6.7.0. Briefly, identifications marked 'Only identified by site,' 'Reverse,' and 'Potential Contaminant' were removed along with identifications made using <2 unique peptides. $Log_2$ transformed LFQ Intensity values were grouped into control and knockout groups, each consisting of three replicates. Proteins not quantified in at least two replicates from each group were removed from the analysis. Annotations (Gene Ontology Biological Process [GOBP]) were loaded through matching with the majority protein ID. A two-sample, two-sided t-test was performed on the values with significance determined using permutation-based FDR statistics (FDR 5%, S0 = 1). Enrichment analysis of GOBP terms was performed on significantly altered proteins using a significance threshold of 4% FDR.

## Statistical analysis

Statistical analyses were conducted using Microsoft Excel and Prism (GraphPad). Statistical significance was determined either by two-tailed Student's t-test, one-way ANOVA using the Bonferroni comparisons test with a 95% confidence interval or nested ANOVA, as indicated in the figure legends. Significance was calculated, where * indicates p<0.05, ** indicates p<0.01, *** indicates p<0.001, and **** indicates p<0.0001.

## Acknowledgements

We would like to thank Markus Kerr, Nicholas Ariotti, Aaron Smith, and Brian Gabrielli for valuable discussion. This work was supported by fellowships and grants from the National Health and Medical Research Council of Australia (to RGP [grants APP1140064 and APP1150083 and fellowship APP1156489], RGP and ASY [grant number APP1037320], ASY [grant number APP1044041], MTR and DAS [grant number APP1125390], and DAS grant numbers [APP1070916 and APP1140851]) as well as by the Australian Research Council Centre of Excellence in Convergent Bio-Nano Science and Technology (RGP) and the Kids Cancer Project of the Oncology Research Foundation (ASY). Confocal microscopy was performed at the Australian Cancer Research Foundation (ACRF)/Institute for Molecular Bioscience (IMB) Dynamic Imaging Facility for Cancer Biology, established with funding from the ACRF. The authors acknowledge the use of the Monash Biomedical Proteomics Facility for the provision of instrumentation, training, and technical support. We also thank Beric Henderson (Westmead Institute for Cancer Research, University of Sydney, Australia) for the BRCA1-YFP construct.

## Additional information

### Funding

| Funder | Grant reference number | Author |
| --- | --- | --- |
| National Health and Medical Research Council | APP1140064 | Robert G Parton |
| National Health and Medical Research Council | APP1150083 | Robert G Parton |
| National Health and Medical Research Council | APP1156489 | Robert G Parton |
| National Health and Medical Research Council | APP1037320 | Robert G Parton |
| National Health and Medical Research Council | APP1044041 | Alpha Yap |
| National Health and Medical Research Council | APP1125390 | Michael T Ryan |
| National Health and Medical Research Council | APP1070916 | David A Stroud |
| National Health and Medical Research Council | APP1140851 | David A Stroud |

The funders had no role in study design, data collection and interpretation, or the decision to submit the work for publication.

### Author contributions

Kerrie-Ann McMahon, Conceptualization, Formal analysis, Writing - original draft, Writing - review and editing; David A Stroud, Michael T Ryan, Formal analysis, Methodology, Writing - review and editing; Yann Gambin, Vikas Tillu, Michele Bastiani, Emma Sierecki, Mark E Polinkovsky, Thomas E Hall, Guillermo A Gomez, Yeping Wu, Marie-Odile Parat, Nick Martel, Harriet P Lo, Data curation, Formal analysis, Writing - review and editing; Kum Kum Khanna, Kirill Alexandrov, Roger Daly, Alpha Yap, Resources, Formal analysis, Writing - review and editing; Robert G Parton, Resources, Data curation, Funding acquisition, Writing - review and editing

### Author ORCIDs

Kerrie-Ann McMahon  https://orcid.org/0000-0002-0833-5708
Yann Gambin  http://orcid.org/0000-0001-7378-8976
Vikas Tillu  http://orcid.org/0000-0002-1034-9543
Thomas E Hall  http://orcid.org/0000-0002-7718-7614
Roger Daly  http://orcid.org/0000-0002-5739-8027
Robert G Parton  https://orcid.org/0000-0002-7494-5248

### Decision letter and Author response

Decision letter https://doi.org/10.7554/eLife.61407.sa1
Author response https://doi.org/10.7554/eLife.61407.sa2

## Additional files

### Supplementary files

• Supplementary file 1. Complete label-free quantitative proteomics for cavin3 KO cells. Complete list of proteins analyzed in cavin3 KO compared to WT HeLa cells (control). Significant ($p<0.05$) mean $\log_2$ transformed SILAC ratios.

• Supplementary file 2. Pathway analysis for cavin3 KO cells. Gene Ontology Biological Process (GOBP) name of both significantly upregulated and downregulated pathways with their corresponding p-values and enrichment scores.

- Supplementary file 3. Supplementary discussion and references.
- Transparent reporting form

## Data availability

All reagents are available from the corresponding author upon request. Proteomics data that supports the findings of this study is presented in Supplementary File 1 and 2. Raw western blots with molecular weight markers are presented in source data files. The raw mass spectrometry proteomics data for this manuscript comparing HeLa WT and HeLa cavin3 KO cells has been deposited to the ProteomeXchange Consortium via the PRIDE partner repository with the dataset identifier PXD026724.

The following dataset was generated:

| Author(s) | Year | Dataset title | Dataset URL | Database and Identifier |
|-----------|------|---------------|-------------|-------------------------|
| Stroud DA | 2021 | Cavin3 released from caveolae interacts with BRCA1 to regulate the cellular stress response | https://www.ebi.ac.uk/pride/archive/projects/PXD026724 | PRIDE, PXD026724 |

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

# Appendix 1

**Appendix 1—key resources table**

| Reagent type (species) or resource | Designation | Source or reference | Identifiers | Additional information |
|---|---|---|---|---|
| Cell line (*Homo sapiens*) | MCF7 cells | ATCC | ATCC: HBT-22 RRID:CVCL_0031 | *Figure 2*, *Figure 3*, *Figure 4*, *Figure 7A–C*, *Figure 3—figure supplement 1*, *Figure 3—figure supplement 2*, *Figure 4—figure supplement 1A–C, E*, *Figure 7—figure supplement 1E* |
| Cell line (*H. sapiens*) | MDA-MB231 cells | ATCC | ATCC: HTB-26 RRID:CVCL_0062 | *Figure 7E, G*, *Figure 2—figure supplement 1*, *Figure 4—figure supplement 1E*, *Figure 4—figure supplement 2*, *Figure 6—figure supplement 1*, *Figure 7, Figure 7—figure supplement 1C, D* |
| Cell line (*H. sapiens*) | A431 cells | ATCC | ATCC:CRL-1555 RRID:CVCL_0037 | *Figure 5*, *Figure 6*, *Figure 7D*, *Figure 4—*, *Figure 4—figure supplement 1D-F*, *Figure 4—figure supplement 3*, *Figure 6—figure supplement 2*, *Figure 7—figure supplement 1A, B* |
| Cell line (*H. sapiens*) | HeLa WT cells | ATCC | ATCC: CRM-CCL-2, RRID:CVCL_0030 | *Figure 1*, *Figure 8*, *Figure 1—figure supplement 1*, *Figure 7—figure supplement 2*, *Figure 8—figure supplements 1* and *2* |
| Cell line (*H. sapiens*) | HeLa cavin3 KO cells | This paper | | *Figure 1*, *Figure 8*, *Figure 1—figure supplement 1*, *Figure 7—figure supplement 2*, *Figure 8—figure supplements 1* and *2* |
| Antibody | 53BP1 rabbit polyclonal | GeneTex | GeneTex Cat# GTX112864 | WB 1:1000 |

*Continued on next page*

*Appendix 1—key resources table continued*

| Reagent type (species) or resource | Designation | Source or reference | Identifiers | Additional information |
|---|---|---|---|---|
| Antibody | ACCA rabbit polyclonal | Cell Signaling | Cell Signaling Cat# 3662 RRID:AB_2219400 | WB 1:5000 |
| Antibody | Actin mouse monoclonal | Millipore | Millipore Cat# MAB1501, RRID: AB_2223041 | WB 1:5000 |
| Antibody | ACLY rabbit polyclonal | Sigma-Aldrich | Sigma-Aldrich Cat# HPA028758, RRID:AB_10603575 | WB 1:2000 |
| Antibody | Aurora kinase mouse monoclonal | BD Biosciences | BD Biosciences Cat# 611082, RRID:AB_2227708 | PLA 1:100 |
| Antibody | Alexa Fluor 488 Goat anti-Rabbit IgG (H + L) | Thermo Fisher Scientific | Thermo Fisher Scientific Cat# A-11034, RRID:AB_2576217 | IF: 1:500 |
| Antibody | Alexa Fluor 546 Goat anti-Mouse IgG (H + L) | Thermo Fisher Scientific | Thermo Fisher Scientific Cat# A-11030, RRID:AB_2534089 | IF 1:500 |
| Antibody | Alexa Fluor 594 Donkey anti-Rabbit IgG (H + L) | Thermo Fisher Scientific | Thermo Fisher Scientific Cat# A-21207, RRID:AB_141637 | IF 1:500 |
| Antibody | Alexa Fluor 594 Goat anti-Mouse IgG (H + L) | Thermo Fisher Scientific | Thermo Fisher Scientific Cat# A-21203, RRID:AB_141633 | IF 1:500 |
| Antibody | BARD1 E-11 mouse monoclonal | Bio-Strategy Laboratory Products | Santa Cruz Cat# sc-74559, RRID:AB_2061237 | WB 1:500 |
| Antibody | BRCA1 C-20 rabbit polyclonal | Bio-Strategy Laboratory Products | Santa Cruz Cat# sc-642, RRID: AB_630944 | WB 1:500 IF 1:100 PLA 1:100 |
| Antibody | BRCA1 MS110 mouse monoclonal | Abcam | Abcam Cat# ab16780, RRID: AB_2259338 | WB 1:1000 IF 1:100 PLA 1:100 |
| Antibody | BRCA1 D-9 mouse monoclonal | Bio-Strategy Laboratory Products | Santa Cruz Cat# sc-6954, RRID: AB_626761 | IF1:50 |
| Antibody | BRCA1 rabbit polyclonal | Millipore | Millipore Cat# 07-434, RRID: AB_2275035 | WB 1:2000 |
| Antibody | BRCA1 rabbit polyclonal | Proteintech | Proteintech Cat# 22363-1-AP, RRID:AB_2879090 | WB 1:1000 |
| Antibody | BRCA2 rabbit polyclonal | BioVision | BioVision Cat# 3675-30T, RRID: AB_2067764 | WB 1:2000 |
| Antibody | BRCC36 rabbit polyclonal | ProScience | ProScience Cat# 4311 | WB 1:1000 |
| Antibody | BRCC45 rabbit polyclonal | GeneTex | GeneTex Cat# GTX105364, RRID:AB_1949757 | WB 1:2000 |
| Antibody | Caldesmon mouse monoclonal | BD Biosciences | BD Biosciences Cat#610660 | WB 1:3000 |
| Antibody | Catenin- alpha mouse monoclonal | Cell Signaling | Cell Signaling Cat# 2131 | WB 1:3000 |
| Antibody | Catenin-gamma mouse monoclonal | Cell Signaling | Cell Signaling Cat# 2309 | WB 1:3000 |
| Antibody | Caveolin1 (CAV1) rabbit polyclonal | BD Biosciences | BD Biosciences Cat#610060, RRID:AB_397472 | WB 1: 5000 |
| Antibody | cavin1 mouse monoclonal | Abmart, China | | PLA 1:100 |
| Antibody | cavin1 rabbit polyclonal | Sigma-Aldrich | Sigma-Aldrich Cat# AV36965, RRID:AB_1855947 | WB 1:2000 |

*Continued on next page*

*Appendix 1—key resources table continued*

| Reagent type (species) or resource | Designation | Source or reference | Identifiers | Additional information |
|---|---|---|---|---|
| Antibody | cavin3 mouse monoclonal | Novus | Novus Cat# HOO112464-MO, RRID:AB_11188730 | PLA 1:200 |
| Antibody | cavin3 rabbit polyclonal | Proteintech | Proteintech Cat# 16250-1-AP, RRID:AB_2171897 | WB 1:2000 IF 1:300 PLA 1:200 |
| Antibody | CHD3 rabbit polyclonal | GeneTex | GeneTex Cat# GTX131779, RRID:AB_2886520 | WB 1:500 |
| Antibody | DDX21 rabbit polyclonal | Novus | Novus Cat# NBP1-88310, RRID:AB_11027665 | WB 1:2000 |
| Antibody | EGFR Clone LA22 mouse monoclonal | Millipore | Millipore Cat# 05-104, RRID: AB_11210086 | WB 1:4000 |
| Antibody | FANCD2 N1 mouse monoclonal | GeneTex | GeneTex Cat# GTX116037, RRID:AB_2036898 | WB 1:500 |
| Antibody | Flotillin1 Clone 18 mouse monoclonal | BD Biosciences | BD Biosciences Cat# 610821, RRID:AB_398140 | PLA 1:100 |
| Antibody | GFP mouse monoclonal | Roche | Roche Cat#11814460001, RRID: AB_390913 | WB 1:4000 PLA 1:300 |
| Antibody | Histone H2.AX-Chip Grade | Abcam | Abcam Cat# ab20669, RRID: AB_445689 | WB 1:1000 |
| Antibody | Histone H2.AX (20E3) P139 | Cell Signaling Technology | Cell Signaling Technology Cat# 9718, RRID:AB_2118009 | IF 1:500 |
| Antibody | Histone H2.AX Chip Grade P139 rabbit polyclonal | Abcam | Abcam Cat# ab2893, RRID:AB_303388 | WB 1:3000 |
| Antibody | HLTF rabbit polyclonal | Proteintech | Proteintech Cat# 14286-1-AP, RRID:AB_2279646 | WB 1:2000 |
| Antibody | HRP-Goat anti-Mouse IgG (H + L) | Thermo Fisher Scientific | Thermo Fisher Scientific Cat# G-21040, RRID:AB_2536527 | WB 1:5000 |
| Antibody | HRP-Goat anti-Rabbit IgG (H + L) | Thermo Fisher Scientific | Thermo Fisher Scientific Cat# G-21234, RRID:AB_2536530 | WB 1:5000 |
| Antibody | HRP-rabbit anti-sheep IgG (H + L) | Abcam | Abcam Cat# ab97130, RRID: AB_10679515 | WB 1:2000 |
| Antibody | MDC1 rabbit polyclonal | Novus | Novus Cat#NB10056657, RRID: AB_838567 | WB 1:100 |
| Antibody | Merit40 sheep polyclonal | R&D Systems | R&D Systems Cat# AF6604, RRID:AB_10717577 | WB 1:500 |
| Antibody | PARP1 rabbit polyclonal | GeneTex | GeneTex Cat# GTX112864, RRID:AB_11173565 | WB 1:1000 |
| Antibody | PCNA mouse monoclonal | Millipore | Millipore Cat# NA03T, RRID: AB_2160357 | PLA: 1:100 |
| Antibody | PGK1 rabbit polyclonal | GeneTex | GeneTex Cat# GTX107614, RRID:AB_2037666 | WB 1:3000 |
| Antibody | PKM rabbit polyclonal | GeneTex | GeneTex Cat# GTX107977, RRID:AB_1951264 | WB 1:3000 |
| Antibody | Rad51 mouse monoclonal | Novus | Novus Cat# NB 100-148, RRID: AB_350083 | WB 1:1000 |
| Antibody | RAP80 D1T6Q rabbit polyclonal | Cell Signaling Technology | Cell Signaling Technology Cat# 14466, RRID:AB_2798487 | WB1:1000 IF 1:100 |

*Continued on next page*

*Appendix 1—key resources table continued*

| Reagent type (species) or resource | Designation | Source or reference | Identifiers | Additional information |
|---|---|---|---|---|
| Antibody | RNF168 rabbit polyclonal | GeneTex | GeneTex Cat# GTX118147, RRID:AB_11169617 | WB 1:1000 |
| Antibody | Tubulin (DM1A) mouse monoclonal | Abcam | Abcam Cat# ab7291, RRID:AB_2241126 | WB 1:4000 |
| Sequence-based reagent | CHD3 human | Integrated DNA Technologies | Hs.Cas9.CHD3.1.AA, strand sequence *GACCGGG TCGGAAACGAAGA* | |
| Sequence-based reagent | FANCD2 human | Integrated DNA Technologies | Hs.Cas9.FANCD2.1.AA, strand sequence *AGTTGACTGACAATGAGTCG* | |
| Sequence-based reagent | PARP1 human | Integrated DNA Technologies | Hs.Cas9.PARP1.1.AA, strand sequence *GAGTCGAG TACGCCAAGAGC* | |
| Sequence-based reagent | 53BP1 human | Integrated DNA Technologies | Hs.Cas9.TP53BP1.1.AA strand sequence *AACGAGGAGACGGTAATAG T* | |
| Sequence-based reagent | siRNAs to BRCA1 human | Life Technologies | HSS101089 HSS186096 HSS186097 | |
| Sequence-based reagent | siRNAs to cavin3 human | Life Technologies | HSS174185 HSS150811 HSS150809 | |
| Commercial assay or kit | Cytotoxicity Detection Kit$^{PLUS}$ LDH | Sigma-Aldrich | Sigma-Aldrich: 4744934001 | |
| Commercial assay or kit | Duolink In situ PLA Probe anti-Rabbit MINUS | Sigma-Aldrich | Sigma-Aldrich Cat# DUO92005, RRID:AB_2810942 | |
| Commercial assay or kit | Duolink In situ PLA Probe anti-Mouse PLUS | Sigma-Aldrich | Sigma-Aldrich Cat# DUO92001, RRID:AB_281039 | |
| Commercial assay or kit | Duolink In situ detection reagent Orange | Sigma-Aldrich | Sigma-Aldrich: DUO92007 | |
| Commercial assay or kit | PrestoBlue Viability Reagent (x10) | Life Technologies | Life Technologies: A13261 | |
| Chemical compound, drug | AZD2461 | Sigma-Aldrich | Sigma- Aldrich: SML 1858 | |
| Chemical compound, drug | CRISPR MAX kit | Life Technologies | Life Technologies: CMAX00001 | |
| Chemical compound, drug | cOmplete, mini EDTA-free protease inhibitor cocktail | Sigma-Aldrich | Sigma-Aldrich: 11836170001 | |
| Chemical compound, drug | DMEM | Gibco/Thermo Fisher | Gibco/Thermo Fisher: 10313-021 | |
| Chemical compound, drug | FBS SERANA | Fisher Biotechnology | Fisher Biotechnology: FBS-AU-015 batch no: 18030416 | |

*Continued on next page*

*Appendix 1—key resources table continued*

| Reagent type (species) or resource | Designation | Source or reference | Identifiers | Additional information |
|---|---|---|---|---|
| Chemical compound, drug | G418 | Sigma-Aldrich | Sigma-Aldrich: 472788001 | |
| Chemical compound, drug | Hydrogen peroxide 3% (w/w) solution | Sigma-Aldrich | Sigma-Aldrich: H1009 | |
| Chemical compound, drug | L-glutamine 100X | Gibco/Thermo Fisher | Gibco/Thermo Fisher: 25030-081 | |
| Chemical compound, drug | Lipofectamine 3000 Reagent | Thermo Fisher | Thermo Fisher: L3000015 | |
| Chemical compound, drug | MG132 (Z-Leu-Leu-Leu-al) | Sigma-Aldrich | Sigma-Aldrich: C2211 | |
| Chemical compound, drug | OptiMem reduced serum medium | Thermo Fisher | Thermo Fisher: 31985070 | |
| Chemical compound, drug | PhosSTOP Phosphatase Inhibitors | Sigma-Aldrich | Sigma-Aldrich: 4906837001 | |
| Chemical compound, drug | Trypsin-EDTA (0.05%) phenol red | Gibco/Thermo Fisher | Gibco/Thermo Fisher: 25300062 | |
| Software, algorithm | GraphPad Prism | GraphPad Prism (https://graphpad.com) | RRID:SCR_015807 | Version 9 |
| Software, algorithm | ImageJ | ImageJ (http://imagej.nih.gov/ij) | RRID:SCR_003070 | |

