## [Decision Letter]

**Acceptance summary:**

In this work, the authors identify a novel interaction between cavin3 and the tumor suppressor BRCA1, unveiling a molecular mechanism for communications between the morphological cellular structures and DNA repair as part of cellular stress responses. Specifically, cavin3 is best understood for its role as an adapter protein found in caveolae, which are surface structures attached the vertebrae plasma membranes.

Using global proteome analysis, cellular, molecular and microscopy techniques, the data uncovers that cavein3 directly interacts with BRCA1 when released from the plasma membrane under cellular stress. cavein3 depletion results in BRCA1 protein instability and as a result DNA damage and cellular PARP inhibitor sensitivity.

**Decision letter after peer review:**

Thank you for submitting your article "Cavin3 released from caveolae interacts with BRCA1 to regulate the cellular stress response" for consideration by *eLife*. Your article has been reviewed by 3 peer reviewers, one of whom is a member of our Board of Reviewing Editors, and the evaluation has been overseen by Jessica Tyler as the Senior Editor. The following individual involved in review of your submission has agreed to reveal their identity: Libin Liu (Reviewer #3).

The reviewers have discussed the reviews with one another and the Reviewing Editor has drafted this decision to help you prepare a revised submission.

Summary:

Using global proteome analysis the authors state to identify a novel tumor-suppressive function for cavin3 mediated through its interaction with BRCA1, leading to regulation of BRCA1 levels, subcellular location, and function. The authors further state that cavin3 controls BRCA1 functions in UV-induced apoptosis and cell protection against DNA damage through downregulation and abolishment of the recruitment of the BRCA1 A-complex to DNA lesions in response to UV damage.

Overall the reviewers concluded that the study shows an interesting interaction between cavin3 and BRCA1, significantly expanding and substantiating previous ancillary reports. However, some of the reviewer's enthusiasm was significantly dampened by the preliminary data on DNA repair, and overstatements and perhaps over-interpretations of the data at several places in the manuscript. There further was concern about overexpression data and the usage of cells that do not contain cavin3, and as a consequence the pathophysiological relevance was brought into question.

Essential revisions:

Currently, the PARPi sensitivity studies are the only data showing a link to DNA repair. However, the data is unconvincing. PARPi does not confer any cell killing for 3-5 cell cycles, and with the amount of BRCA1 destabilization a most dramatic sensitivity would be expected (e.g. low nM ranges, not μm ranges). This suggests the possibility that the observed differences in cell survival may be due to non-specific toxicity. Moreover, it is not clear why there is a loss of gH2AX, since loss of BRCA1 would be expected to increase gH2AX, rather than decrease. This could hint to more global structural issues with the loss of cavin3, such as actin interactions disturbing repair sites as a whole, rather than direct disruption of BRCA1-cavin3 interactions. The data suggesting any claim of affecting DNA repair should be substantiated with additional data, such as Dr-GFP assays and/or comet assays. Since many DNA repair proteins appear to be lost with cavin3, the dependence of the physiological outcomes on BRCA1 as portrayed needs to be substantiated, e.g. by rescue experiments with concomitant knock-out of 53BP1, which restores BRCA1 (but not BRCA2) dependent sensitivity to PARPi.

The overall claim is that cavin3's potential tumor suppressor function is through a direct interaction with BRCA1 at its N-terminus. Since many DNA repair proteins are lost with cavin3 (including RAD51 which in other systems would not be affected by loss of BRCA1), can the authors exclude interactions with other DNA repair proteins, e.g. by PLA between cavin3 and RAD51?

---

## [Author Response]

Essential revisions:

Currently, the PARPi sensitivity studies are the only data showing a link to DNA repair. However, the data is unconvincing. PARPi does not confer any cell killing for 3-5 cell cycles, and with the amount of BRCA1 destabilization a most dramatic sensitivity would be expected (e.g. low nM ranges, not μm ranges). This suggests the possibility that the observed differences in cell survival may be due to non-specific toxicity.

We have now performed clonogenic and prestoblue viability assays in WT and cavin3 KO cells treated with the PARP inhibitor at low nM concentrations for 6 days in Figure 8—figure supplement 2A-C. Clonogenic survival and cell viability studies revealed that cavin3-deficient HeLa cells were significantly more sensitive to PARP inhibitor at 5 nM concentration than control WT HeLa cells (see Figure 8—figure supplement 2A-C, black and red dots). These results suggest that cavin3-deficient cells have reduced homologous recombination DNA repair similar to BRCA1-deficient cells.

We have further substantiated these findings by generating CRISPR-Cas9 PARP1 KO in both WT and cavin3 KO cells. Cavin3/PARP1 KO cells exhibit a profound loss of colonies in clonogenic assays and a significant decrease in prestoblue viability assays as compared to WT or PARP1 KO cells (see Figure 8—figure supplement 2A-C, blue dots). Collectively these findings suggest that cavin3-deficient HeLa cells are sensitised to PARP inhibition suggesting involvement of cavin3 in homologous recombination DNA repair (also see new results as outlined below).

Moreover, it is not clear why there is a loss of gH2AX, since loss of BRCA1 would be expected to increase gH2AX, rather than decrease. This could hint to more global structural issues with the loss of cavin3, such as actin interactions disturbing repair sites as a whole, rather than direct disruption of BRCA1-cavin3 interactions. The data suggesting any claim of affecting DNA repair should be substantiated with additional data, such as Dr-GFP assays and/or comet assays.

We have now included comet assays in WT and cavin3 KO cells treated with a low concentration of PARP inhibitor (5 nM AZD 2461) for 6 days (revised Figure 8F). Consistent with the colony formation assays described above, we now show that cavin3 KO cells treated with low nM PARP inhibitor have significantly higher levels of overall DNA damage than equivalent WT HeLa cells. We believe that this strengthens the conclusion that loss of cavin3 contributes to DNA damage and that this can be compensated by functional PARP1.

Since many DNA repair proteins appear to be lost with cavin3, the dependence of the physiological outcomes on BRCA1 as portrayed needs to be substantiated, e.g. by rescue experiments with concomitant knock-out of 53BP1, which restores BRCA1 (but not BRCA2) dependent sensitivity to PARPi.

In Supplementary Figure 13, we have generated CRISPR-Cas9 KO of 53BP1 in WT and cavin3 KO cells to determine the dependence of the physiological outcomes on BRCA1 in cavin3 KO cells. These cells were subjected to clonogenic and prestoblue viability assays with and without PARP inhibitor treatment. We observed that KO of 53BP1 specifically in cavin3 KO cells was able to revert the PARP sensitivity of these cells presumably by restoring BRCA1 dependent homologous recombination repair (see Figure 8—figure supplement 2A-C, orange dots). These findings are in agreement with several studies demonstrating that homologous recombination is partially restored in BRCA1-deficient cells following 53BP1 loss (Cao et al., 2009; Bouwan et al., 2010, Bunting et al., 2009, Turner et al., 2007, Yang et al., 2017). This is now discussed on page 12 of the Results section starting at line 366-377.

The overall claim is that cavin3's potential tumor suppressor function is through a direct interaction with BRCA1 at its N-terminus. Since many DNA repair proteins are lost with cavin3 (including RAD51 which in other systems would not be affected by loss of BRCA1), can the authors exclude interactions with other DNA repair proteins, e.g. by PLA between cavin3 and RAD51?

This is a very interesting suggestion. Since so many DNA repair proteins are lost in our cavin3 KO cells, we have chosen to selectively deplete these cells of several proteins including Chromodomain helicase DNA containing protein 3 (CHD3, an epigenetic modulator) and Fanconi anemia (FA) complementation Group 2 (FANCD2, a DNA damage sensor protein). These proteins were specifically upregulated in cavin3 KO cells and are involved in different aspects of DNA repair. These proteins represent potential targets and mediators of synthetic lethality in cancer and allow us to begin to dissect that role of the cavin3-BRCA1 interaction in different aspects of DNA repair. In Supplementary Figure 13, depletion of CHD3 and FANCD2 specifically in cavin3 KO cells induced a profound decrease in clonogenic survival and cell viability (see Figure 8—figure supplement 2A-C). These findings suggest that in addition to PARP1, CHD3 and FANCD2 are potential synthetic lethal partners for cavin3. These findings also allow us to begin to dissect the role(s) of BRCA1-cavin3 in different aspects of DNA repair and how these proteins function to allow cell survival in stress situations.